# High-Frequency and Ultra-High-Frequency Ultrasound in Dermatologic Diseases and Aesthetic Medicine

**DOI:** 10.3390/medicina61020220

**Published:** 2025-01-26

**Authors:** Giulio Argalia, Alfonso Reginelli, Elisa Molinelli, Anna Russo, Alessandra Michelucci, Andrea Sechi, Angelo Valerio Marzano, Stella Desyatnikova, Marco Fogante, Vittorio Patanè, Giammarco Granieri, Corrado Tagliati, Giulio Rizzetto, Edoardo De Simoni, Marco Matteucci, Matteo Candelora, Cecilia Lanza, Claudio Ventura, Nicola Carboni, Roberto Esposito, Stefano Esposito, Massimiliano Paolinelli, Elisabetta Esposto, Giuseppe Lanni, Gabriella Lucidi Pressanti, Chiara Giorgi, Fabiola Principi, Alberto Rebonato, Sylwia Patrycja Malinowska, Robert Krzysztof Mlosek, Gian Marco Giuseppetti, Valentina Dini, Marco Romanelli, Annamaria Offidani, Salvatore Cappabianca, Ximena Wortsman, Oriana Simonetti

**Affiliations:** 1Maternal-Child, Senological, Cardiological Radiology and Outpatient Ultrasound, Department of Radiological Sciences, University Hospital of Marche, Via Conca 71, 60126 Ancona, Italy; 2Department of Precision Medicine, University of Campania Luigi Vanvitelli, Piazza Luigi Miraglia 2, 80131 Naples, Italy; 3Department of Clinical and Molecular Sciences, Dermatology Clinic, Polytechnic Marche University, Via Conca 71, 60126 Ancona, Italy; 4Department of Dermatology, University of Pisa, Via Roma 67, 56126 Pisa, Italy; 5Interdisciplinary Center of Health Science, Sant’Anna School of Advanced Studies of Pisa, Piazza Martiri della Libertà 33, 56127 Pisa, Italy; 6Dermatology Unit, Fondazione IRCCS Ca’ Granda Ospedale Maggiore Policlinico, Via Pace 9, 20122 Milan, Italy; 7Department of Pathophysiology and Transplantation, Università degli Studi di Milano, Via Francesco Sforza 35, 20122 Milan, Italy; 8The Stella Center for Facial Plastic Surgery, 509 Olive Way Ste 1430, Seattle, WA 98101, USA; 9AST Ancona, Ospedale di Comunità Maria Montessori di Chiaravalle, Via Fratelli Rosselli 176, 60033 Chiaravalle, Italy; 10Gemini Med Diagnostic Clinic, via Tabellione 1, 47891 Falciano, San Marino; 11Poliambulatorio CPV, Via Vivaldi 2-4, 60025 Loreto, Italy; 12AST Ancona, Distretto Sanitario di Senigallia, Dermatologia, Via Campo Boario 4, 60019 Senigallia, Italy; 13AST Pesaro-Urbino, Distretto Sanitario di Pesaro, Via XI Febbraio, 61121 Pesaro, Italy; 14Department of Services, U.O.S.D. Radiology, San Liberatore Hospital, Viale Risorgimento, 64032 Atri, Italy; 15AST Pesaro-Urbino, Radiologia, Ospedale Santa Maria della Misericordia, Via Comandino 70, 61029 Urbino, Italy; 16AST Ancona, Radiologia, Ospedale Santa Casa di Loreto, Via San Francesco 1, 60025 Loreto, Italy; 17AST Pesaro-Urbino, Radiologia, Ospedale San Salvatore, Piazzale Cinnelli 1, 61121 Pesaro, Italy; 18Life-Beauty—Private Company, Ul. T. Kosciuszki 29, 05-825 Grodzisk Mazowiecki, Poland; 19Diagnostic Ultrasound Laboratory, Medical University of Warsaw, 61 Zwirki i Wigury Street, 02-091 Warszawa, Poland; 20Medical Point Ancona, Via Trieste 21, 60124 Ancona, Italy; 21Department of Dermatology, Faculty of Medicine, Universidad de Chile, Lo Fontecilla 201 of 734 Las Condes, Región Metropolitana de Santiago, Santiago 8330111, Chile; 22Department of Dermatology, School of Medicine, Pontificia Universidad Catolica de Chile, Av. Libertador Bernardo O’Higgins 340, Región Metropolitana de Santiago, Santiago 8331150, Chile; 23Institute for Diagnostic, Imaging and Research of the Skin and Soft Tissues (IDIEP), Lo Fontecilla 201 of 734 Las Condes, Región Metropolitana de Santiago, Santiago 7591018, Chile; 24Department of Dermatology and Cutaneous Surgery, Miller School of Medicine, University of Miami, 1120 NW 14th St Ste 9, Miami, FL 33146, USA

**Keywords:** dermatology, high-frequency ultrasound, ultra-high-frequency ultrasound

## Abstract

Dermatologic ultrasonography applications are rapidly growing in all skin fields. Thanks to very high spatial resolution, high-frequency and ultra-high-frequency ultrasound can evaluate smaller structures, allowing us to improve diagnosis accuracy and disease activity. Moreover, they can guide treatment, such as drug injection, and assess therapy efficacy and complications. In this narrative review, we evaluated high-frequency ultrasound and ultra-high-frequency ultrasound in infections, inflammatory dermatoses, metabolic and genetic disorders, specific cutaneous structure skin disorders, vascular and external-agent-associated disorders, neoplastic diseases, and aesthetics.

## 1. Introduction

Many imaging modalities have been used in dermatology, some for a long time, such as dermoscopy, and others are newer, such as optical coherence tomography and reflectance confocal microscopy [1,2].

Radiologists can evaluate skin and skin-related disorders through ultrasound, X-ray, computed tomography, magnetic resonance imaging, and positron emission tomography.

The imaging modality that both radiologists and dermatologists perform is ultrasound. New advancements such as high and ultra-high frequency ultrasound allow us to better evaluate skin layers and diseases.

Linear probes with frequencies of 15 MHz or greater are needed to evaluate skin layers. High-frequency ultrasound (HFUS) is usually considered when a 20–30 MHz transducer is used, allowing tissue penetration of about 10 mm, which is useful for dermis and subcutaneous tissue evaluation. Ultra-high-frequency ultrasound (UHFUS) is defined as when probes with higher frequencies are used, with a penetration depth of about 3 and 4 mm, respectively, for 48 MHz and 70 MHz probes, which allow a better evaluation of epidermis, hair follicles, nail units, and vessels, including lymphatics [3] (Figure 1).

In this narrative review, we extensively evaluated HFUS and UHFUS in infections, inflammatory dermatoses, metabolic and genetic disorders, specific cutaneous structure skin disorders, vascular and external-agent-associated disorders, neoplastic diseases, and aesthetics.

## 2. Infections

Ultrasound is useful in an emergency department setting to differentiate between abscesses and other skin and soft tissue infections such as cellulitis; this is very important, as abscesses can be managed using drainage, whereas cellulitis can be treated with antibiotics [4]. This differentiation can also be reached in the pediatric population, in which a rapid, non-invasive, painless, easily repeatable ultrasound examination can allow for a change in the management of an equivocal physical examination [5]. Abscesses are irregular, hypoechoic, or anechoic lesions with variable echogenic debris, surrounded by a hyperechoic rim and peripheral hypervascularity; cellulitis can show diffuse dermis and subcutis thickening with a hyperechoic “cobblestone” appearance and increased vascularity (Figure 2, Figure 3 and Figure 4) [6].

In cases in which clinical examination or dermoscopy are non-conclusive for scabies, HFUS can help identify the mites as hyperechoic, millimetric structures within the epidermis or at the epidermal–dermal junction and their burrows as linear or curvilinear hypoechoic tracks in the stratum corneum; moreover, HFUS can help in assessing treatment accuracy and efficacy, as the presence or absence of mites within the epidermis of non-specific burrow-like scratches can be evaluate in patients with persistent pruritus [7].

Warts are caused by the human papillomavirus and appear on ultrasound as focal fusiform hypoechoic epidermal and dermal structures, with irregular epidermis and dermal thickening and commonly hypervascularity on Doppler examination; ultrasound can help monitor treatment responses, particularly in recurrent and difficult cases with persistent pain [8].

HFUS allows the evaluation of odontogenic cutaneous sinus tract and plays a crucial diagnostic role, as frequently it is not clinically suspected; in fact, a nodular, triangular, or mushroom-shaped lesion can be detected in the subcutis, which continues with a slightly tortuous band structure, up to the focally interrupted cortical bone plate; hypervascularity around and/or within the lesion due to inflammation and a hard strain elastography pattern of the sinus tract due to fibrous and granulomatous tissue are reported [9].

Moreover, HFUS can be used in other skin infections to better evaluate epidermis, dermis, and subcutis and their alterations, such as epidermal thickening, dermal disorganization, dermal decreased echogenicity, subcutis hyperechogenicity, and increased vascularization in chromoblastomycosis [10].

HFUS is reported to be very useful in the diagnosis, evaluation of deep hypodermal involvement, intralesional ultrasound-guided drug infiltration, and follow-up of cutaneous leishmaniasis; moreover, ultrasound can help differentiate two disease patterns, inflammatory and pauci-inflammatory ones, with shorter healing time for lesions with a pauci-inflammatory pattern (Figure 5); in fact, inflammatory patterns showed central vascularity, ill-defined margins, and more signs of panniculitis than pauci-inflammatory ones [11,12] (Figure 6).

HFUS is reported to be the best imaging technique for the evaluation of nerve disease in leprosy, particularly for the assessment of ulnar, median, common fibular, and tibial neuropathy in Hansen’s disease without evident skin involvement, as leprosy primarily involves Schwann cells and secondarily affects the skin and mucous membrane with anesthetic patches or deformities; when a pure neuritic Hansen’s disease is detected when assessing peripheral nerve cross-sectional area, immediate treatment is necessary to prevent complications as, if the skin is already involved, it is usually too late to prevent them; moreover, asymmetric multiple nerve thickening or asymmetry could indicate evidence of Hansen’s disease neuropathy among household contacts of patients with leprosy, and this could become a useful tool for an early diagnosis of leprosy cases [13,14,15].

HFUS and UHFUS can help diagnose and monitor the treatment and better understand the spreading mechanisms of cutaneous larva migrans, as the larva can be identified as a small linear hyperechoic subepidermal structure, the dilated lymphatic ducts can be detected as hypoechoic dermal and hypodermal tunnels, inflammation can be revealed by hypoechoic dermis, hyperechoic subcutis, and increased vascularity on color Doppler imaging [16]. Fly larvae (myasis) can cause skin infestation, and on HFUS central hyperechoic moving structures can be shown, with a hypoechoic rim and peripheral vascularity on color Doppler imaging [17].

Actinomycetomas and eumycetomas can show hypoechoic areas or connecting tracts with increased rim vascularity [18].

In patients with gonorrhea, ultrasound can demonstrate paraurethral duct diameter and length before and after treatment and can guide incision, drainage, and wedge resection when antibiotics are not sufficient [19,20].

## 3. Inflammatory Dermatoses

Plaque psoriasis shows increased thickness of the epidermis and dermis, with frequent epidermal thickening and undulation with dermal hypoechogenicity due to inflammation and a lack of subcutis abnormalities [21]

Lichen planus papules can demonstrate increased thickness and a decreased echogenicity of the upper dermis, with hypoechoic dermis, related to epidermal acanthosis and dermal inflammatory infiltrate [22,23].

The upper dermal hypoechoic band thickness significantly decreases during dupilumab treatment, and it shows a high correlation with clinical scores in patients with moderate or severe atopic dermatitis [24,25,26] (Figure 7 and Figure 8).

Epidermial irregularity and hyperechogenicity, a hypoechoic linear band at the dermo-epidermal junction, and modest dermal thickening were found in seborrheic dermatitis [27].

In cutaneous chronic graft-versus-host disease, HFUS can show epidermis and subcutis thinning and dermis thickening compared to healthy controls; moreover, HFUS could help obtain an earlier diagnosis of this disease [28].

A widened and hypoechoic dermis, hypoechoic dermal areas, hypoanechoic fusiform structures below the epidermis representing localized infiltrate, cutaneous and subcutaneous edema with superficial thrombophlebitis, and chronic thrombosis with punctate intraluminal calcifications are possible imaging findings in cutaneous mastocytosis [29,30].

UHFUS in pyoderma gangrenosum can show hyperechoic oval structures and hair tracts in the inflammatory phase, and the identification of these early inflammatory imaging signs could, in the future, let dermatologists avoid biopsies to diagnose this disease [31] (Figure 9).

UHFUS can accurately assess epidermal, dermal, and subcutis thickness and precisely identify the epidermal–dermal junction and the boundary between the dermis and subcutaneous tissue. Therefore, it could be useful to differentiate blister locations, such as subepidermal and intraepidermal bullae, respectively, in bullous pemphigoid and pemphigus vulgaris [32] (Figure 10 and Figure 11). Moreover, intraoral UHFUS can help differentiate between oral pemphigus vulgaris and mucous membrane pemphigoid, as statistically significant differences between the echogenicity of these two lesions were identified [33].

Ultrasound can present vascular density reduction in the fingertip with a higher resistive index due to vascular wall damage in lupus erythematosus patients, and UHFUS can show hypoechoic zones at dermo-epidermal junctions and superficial dermis in neonatal lupus erythematosus annular erythema due to inflammatory cell infiltration [34,35].

Lupus panniculitis can be detected as a subcutis mild hyperechoic ovoidal pseudo-mass with hypoechoic margin and increased vascularization on color Doppler imaging [34]. Connective tissue panniculitis is a rare cutaneous manifestation of connective tissue diseases associated with lupus erythematosus and dermatomyositis, and HFUS can help reach an early diagnosis in cases with unclear skin symptoms; moreover, HFUS can allow non-invasive clinical decision making, as active connective tissue panniculitis is correlated to the hypoechoic dermis, undefined epidermal–dermal borders, hypodermal thickening and hypoechoic septa, hyperechoic lobules, the presence of a vessel diameter of more than 1 mm, systolic peak > 10 cm/s, and resistance index > 0.7 MHz on color Doppler imaging [36].

Ultrasonography can allow us to discriminate mostly lobular from mostly septal panniculitis. This is relevant since lobular and septal panniculitis are usually due to different entities [37] (Figure 12).

In systemic lupus erythematosus patients, a higher intima media thickness was found in various arterial territories than in healthy controls, with a medium echogenic appearance due to possible inflammation or early atherosclerosis [38]. Moreover, HFUS showed a thicker carotid intima and a thinner carotid media in premenopausal women with systemic lupus erythematosus compared to age-matched healthy controls [39]. US can also detect early articular and periarticular disease, such as joint effusion and synovitis, and in rhupus syndrome cortical erosions can be detected [40].

Ultrasound can be helpful to diagnose rheumatoid nodules in finger tendons as a screening procedure in rheumatoid arthritis patients [41].

UHFUS presented epidermal hypoechogenicity and thickening, as well as dermal hypoechogenicity, in systemic sclerosis patients compared to healthy controls [42]. HFUS of the skin between the nail fold and the distal interphalangeal joint can show skin thickening that is higher than in healthy controls or patients with primary Raynaud’s phenomenon [43]. Moreover, UHFUS can help assess digital artery vasculopathy as all three finger arterial layers were significantly thicker in systemic sclerosis patients than in healthy controls [44].

Inflammatory phase morphea shows dermal thickening and hypoechogenicity with increased subcutaneous vascularity and echogenicity, causing dermal–hypodermal blurredness. In the sclerotic phase, there is dermal thinning and dermal hyperechogenicity and subcutaneous thinning and hyperechogenicity; moreover, color Doppler ultrasound can frequently detect subclinical activity in areas adjacent to the clinically active lesion, in non-adjacent regions, and at clinically inactive lesion sites; in fact, ultrasound morphea activity scoring can better support a systematical description of the cutaneous abnormalities and activity tracking accuracy and evaluate treatment response [45,46,47,48,49,50,51]; ultrasound allows early diagnosis of morphea which is essential to avoid serious consequences of atrophy and fibrosis, as well as after a trigger such as liposuction [52].

The burden of cutaneous sarcoidosis can be objectively evaluated using UHFUS and this new non-invasive disease severity measure correlates with histopathology and clinical evaluation, potentially obviating subjective clinical assessment and the necessity of biopsies [53,54].

## 4. Metabolic and Genetic Disorders

Primary localized cutaneous nodular amyloidosis can show subcutis hypoechoic lesions with linear calcifications, and it should be considered among the diseases that can show dystrophic calcifications, such as autoimmune connective tissue diseases, scleroderma, and cutaneous lupus erythematosus [55].

Scleredema diabeticorum can present evident dermal thickening with hyperechoic foci and reduced visualization of the deep dermis [56].

Ultrasonography can be applied to better evaluate, follow-up, and detect early changes in localized pretibial myxoedema in Graves’ disease, in which a hypoechoic thickened dermis and subcutis with ill-defined dermis–hypodermis boundaries can be observed [57].

Ultrasound can allow us to diagnose or confirm the presence of clinically suspected calcinosis cutis (Figure 13), to guide intralesional drug injection such as sodium thiosulfate, and to evaluate treatment response [58,59].

Ultrasound can be used to evaluate diabetic foot ulcers’ healing quantitatively, and UHFUS of the big toe showed upper dermal hypoechogenicity and thickening, probably due to papillary dermal edema; moreover, epidermis was reported to be thinner in patients with diabetic neuropathy and previous ulcers [60,61,62].

In patients with X-linked dominant protoporphyria, HFUS of the nasal dorsum showed epidermal thickening and dermal hypoechogenicity [63].

Plantar foot calluses in patients with pachyonychia congenita showed an anechoic structure between the epidermis and dermis due to subepidermal blister fluid; this characteristic was not reported in patients with other forms of palmoplantar keratoderma, such as mal de Meleda and epidermolytic palmoplantar keratoderma [64].

Pseudoxanthoma elasticum skin lesions are oval homogeneous hypoechogenic areas in the mid- and deep dermis related to the presence of glycosaminoglycans and a high level of hydration of connective tissue; moreover, undulating skin surfaces with normal epidermis and dermal/epidermal interface are reported, without a significant skin echogenic structure due to the small size of the dermal elastic fiber calcifications [65] (Figure 14).

## 5. Specific Cutaneous Structure and Sites of Skin Disorders

HFUS can evaluate different forms and phases of primary cicatricial alopecia; in fact, the inactive phase showed a lower number or a lack of follicular structures compared to the active phase; active discoid lupus erythematosus showed widened follicular structures with hypoechogenic bands; lichen planopilaris and frontal fibrosing alopecia showed cigar-like shaped follicular structures in the active phase, whereas the inactive phase exhibited irregular and saw-like dermal–subepidermal boundary [66].

UHFUS can evaluate and differentiate between active, inactive, and regrowth phases of alopecia areata; therefore, HFUS can be very useful for disease prognosis [67,68]. The three most common ultrasound features are empty hair follicles, small ovoid hair follicles, and subcutis perifollicular hyperechogenicity [69].

Small ovoid hair follicles and miniaturized hairs were also detectable in female pattern hair loss and senescent alopecia. Active lichen planopilaris and frontal fibrosing alopecia lesions showed perifollicular hypoechogenicity in the mid-dermis and distal ambiguity of hair follicles. Homogeneous dermis without structures was detected in lichen planopilaris and frontal fibrosing alopecia scarring lesions, and as well as in central centrifugal cicatricial alopecia. In the latter, a few remnants of hair follicles are visible, too. Areas of homogeneous dermis without structures are visible in traction alopecia at the cicatricial stage but with intact residual hairs, which helps to rule out other cicatricial alopecia [69].

In dissecting cellulitis subcutis, echogenicity with a clear border outlines the inflamed area and can guide an intralesional steroid injection [69]. This disease presents similar findings to hidradenitis suppurativa, such as dilation of the hair follicles, anechoic or hypoechoic pseudocysts, fluid collections, and tunnels (also called fistulas) [70].

Folliculitis decalvans shows fusion of hair follicles, decreased dermal echogenicity, and increased subcutaneous echogenicity [69].

Ultrasound can image trichilemmal cysts as dermal and/or subcutaneous oval-shaped heterogeneous hypoechoic lesions with inner echogenic material without a tract toward the epidermis, occasionally with a few fragments of hair tracts and calcifications (Figure 15) [71].

Folliculotropic mycosis fungoides shows skin thickening, a prominent upper dermis hypoechoic band due to lymphocytic infiltrates, and some hyperechoic deposits around hair follicles because of mucin degeneration [72].

Scalp metastases are usually hypoechoic solid lesions with increased vascularity, and HFUS and UHFUS can rapidly and cost-effectively assess their size and surrounding anatomy; moreover, ultrasound can aid in directing subsequent imaging and treatment strategies, such as tissue sampling or surgical intervention [73].

Cutis verticis gyrata shows undulation of the cutaneous layers, dermal and hypodermal thickening, corresponding with the elevated clinical zones, followed by folds with normal cutaneous thicknesses [74].

UHFUS of vitiligo presents undulation of the epidermis, hypoechoic subepidermal thin plaques, hypoechoic and thickened regional hair follicles and/or pilosebaceous units with prominent sebaceous glands, as well as regional hypervascularity [75].

UHFUS can be used to assess the efficacy of a long-pulsed alexandrite laser for keratosis pilaris. In fact, four weeks after the fourth treatment, UHFUS demonstrated the flattening of the epidermal bulges, which were consistent with dermoscopic and histologic examinations [76].

HFUS can show pseudocysts, folliculitis, fistulas, calcinosis, and scars in acne patients and can help evaluate disease severity through the SOS-Acne (Figure 16). HFUS can identify acne scar depth and scar width (Figure 17); moreover, HFUS can assess treatment efficacy, such as Er:Yag laser therapy, after which a reduction in scar depth and epidermal thickness are reported. Moreover, CO_2_ fractional resurfacing laser treatment can be evaluated with HFUS, and dermal thickness measurement is suggested before treatment as a thick dermis seems less effectively treated than a thin one [77,78,79,80].

Ultrasound helped prove the primary follicular involvement in hidradenitis suppurativa and allows us to assess subclinical disease and anatomical abnormalities that are clinically unrecognizable; therefore, baseline and follow-up ultrasound examinations are recommended. UHFUS can detect drop-shaped hair follicles, microtunnels, and microcysts. HFUS can adequately evaluate fluid collections, abscesses, pseudocysts, and the location and morphology of tunnels, allowing to correctly assess staging of severity (mSOS-HS) and scoring of activity (US-HSA), therefore assisting physicians in choosing the correct medical or surgical intervention and helping to predict treatment response, usually with low patient discomfort (Figure 18, Figure 19, Figure 20, Figure 21 and Figure 22) [81,82,83]. Moreover, UHFUS can guide intralesional steroid injection in HS flares that do not respond to topical treatment, and UHFUS can be used to assess therapeutic response [84,85]. Presurgical lesion mapping by UHFUS is strongly suggested in moderate and severe HS refractory to previous medical and surgical therapies, as UHFUS enables detailed assessment of lesion extension, particularly identifying tunnels and fistulas, facilitating surgical planning [86,87]. Ultrasound can also evaluate associated or linked diseases, such as acne vulgaris, pilonidal sinus, dissecting cellulitis of the scalp, and perianal infective/inflammatory disease, and their treatment response (Figure 23) [88,89,90,91,92,93].

HFUS can help evaluate nail unit and nail plate thickness functions and diseases [94,95].

Nail psoriasis can show thickened, hyperechoic, and wavy nail plates, with nail bed thickening and hypoechogenicity, focal hypoechoic deposits, and loss of definition in the ventral plate [96,97,98,99]; concomitant enthesopathy of the digital extensor tendon in the distal interphalangeal joint can be evaluated [100]. UHFUS can assess a decrease in the nail plate thickness after one month from the beginning of biologic treatment with mAb anti-TNF-alpha and anti-IL, even before clinical changes are detectable [101] (Figure 24).

Ultrasound can detect nail bed thickening, irregular thickening, and fusion of the nail plates in onychomycosis, and it can display periungual fold thickening, with areas of increased and decreased echogenicity, and increased vascularization in paronychia [102].

Onychopapilloma can be evaluated using HFUS, and matrix involvement evident on ultrasound imaging can help predict postsurgical recurrence. Onychopapilloma can present as a nail bed hypoechoic band, nail plate thickening, hyperechoic focal spots, upward displacement, and irregularities of the ventral plate [103].

Onychomadesis and retronychia can be evaluated well with HFUS, which can help identify defects under the proximal nail fold. Frequently, these conditions can be together. Onychomadesis is the fragmentation of the nail plate, and in retronychia a fragmented nail plate is commonly identified beneath the proximal nail fold. Signs of retronychia included a decreased distance between the origin of the nail plate and the base of the distal phalanx, thickening of the proximal nail fold, and a hypoechoic halo surrounding the origin of the nail plate; moreover, color Doppler can detect hypervascularity areas and help decide if onychectomy is necessary [104] (Figure 25).

Nail lichen planus diagnosis can be supported by HFUS, avoiding potential permanent scars derived from nail biopsies; in fact, thickening, decreased echogenicity, and hypervascularity of the nail bed can be detected, as well as a hypoechoic halo surrounding the origin of the nail plate in all or almost all fingers as well as thickening and decreased echogenicity of the periungual dermis; moreover, ultrasound can help monitor this disease’s treatment [105].

HFUS can diagnose a subungueal glomus tumor, a benign tumor derived from the neuromyoarterial plexus. On ultrasound, it presents as an oval-shaped hypoechoic lesion with well-defined borders, intralesional hypervascularity on color Doppler, and bone cup-scalloping without cortical erosion [106] (Figure 26).

Ultrasound can be used to confirm the presence of a digital myxoid cyst (synovial cyst) and it is helpful to guide steroid injection and to follow up lesion evolution after treatment; a large digital myxoid cyst volume is associated with reduced prolonged cure rate, as well as the presence of osteophytes, older age, and long disease duration; these lesion characteristics can be evaluated to decide when to offer surgical excision [107,108,109] (Figure 27).

Telangiectatic granuloma is a benign vascular tumor that usually appears as a hypoechoic, ill-defined, hypervascular dermal structure at the proximal nail fold without erosion of the nail plate or distal phalanx bony margin [110].

Squamous cell carcinoma is the most common nail apparatus malignancy and is a hypoechoic, hypervascular lesion with ill-defined margins, which causes nail plate and bony margin erosions [110].

In patients with longstanding melanonychia, ultrasound can help detect melanoma as ill-defined, hypoechoic, heterogeneous, and hypervascular structures or masses in the nail bed or matrix with nail plate and bone margin erosions [110].

In auricular and nasal non-melanoma skin cancer, preoperative HFUS can show deep cartilaginous or bone infiltration, thus resulting in radioimmunological treatment rather than surgery; in case of local cartilage infiltration, a more aggressive surgical procedure is performed, reducing the need for a second surgical intervention; when no depth infiltration is identified on ultrasound, surgical excision is the treatment of choice [111].

Intraoral HFUS and UHFUS could help diagnose different oral pathologies and evaluate the depth of invasion and tumor thickness of persisting oral soft tissue lesions to predict adverse histological features preoperatively [112,113].

In the case of congenital exclusively lingual pigmentation, epidermal choristoma can be differentiated by congenital lingual melanotic macules thanks to the demonstration on UHFUS of multiple hyperechoic oval-shaped submucosal structures, consistent with sebaceous glands, without hypervascularity [114].

## 6. Vascular Disorders

HFUS can be used to confirm capillary malformation diagnosis, follow up the lesion, assist laser treatment, and evaluate treatment results. Moreover, HFUS can be useful to diagnose associated vascular malformations, such as arteriovenous malformations [115,116] (Figure 28). Moreover, ultrasound can accurately and reliably diagnose intraneural vascular anomalies of peripheral nerves [117].

HFUS hypoechoic upper dermal hypoechogenicity thickness measurement can objectively assess edema severity and treatment adequacy in patients with venous ulcers, as it showed a significant reduction after compression therapy [118] (Figure 29).

High-frequency color Doppler ultrasound can help detect perforators in deep adipose layers before harvesting super-thin anterolateral thigh flaps for foot and ankle skin and soft tissue defect treatment after traffic accidents, heavy object crush, and mechanical and heat crush injuries [119].

UHFUS allows us to detect more lymphatic vessels than conventional linear probe ultrasound, including lymphatics with diameters smaller than 0.3 mm that can show fluids moving inside the lymphatic lumen, and may detect functioning valves. Preoperatively identifying functional lymphatic vessels is very important for lymphedema cases, as lymphaticovenular anastomosis is an effective and minimally invasive surgical treatment for refractory lymphedema [120]. In fact, UHFUS can be used to identify patient-oriented incisions for lymphaticovenular anastomosis in advanced-stage lymphedema as it helps to detect functioning lymphatic channels where it would have been considered impossible before [121,122].

## 7. External-Agent-Associated Disorders

HFUS can non-invasively and objectively assess skin changes during radiotherapy [123,124].

HFUS can evaluate epidermal and dermis thickness, epidermal edema, dermal fibrosis, and follicle density in traumatic scars [125].

Ultrasound can be used to assess surgical wounds, as peri-incisional fluid collection detection is significantly associated with surgical site infection, suggesting a need for early therapy [126,127].

HFUS can help distinguish between scarring, hypertrophic scars, and keloids, as hypertrophic scars show a striped hypoechoic area in the upper dermis. Keloids are characterized by a focal hypoechoic band with a laminar pattern extending beyond the scar’s original site. In atrophic scars, the edges between the dermis and subcutis are not well defined [77] (Figure 30).

HFUS can be used for leg ulcer imaging and healing process monitoring, such as after laser biostimulation of shin ulcers [128].

Ultrasound can detect radiolucent foreign bodies in extremities with high accuracy as hyperechoic structures, and UHFUS is a valuable tool for hand surgery; inert foreign bodies such as metal or glass generate a posterior acoustic reverberation; moreover, inflammation and granulomatous tissue in the periphery of a foreign body determine the frequent visualization of hypoechoic tissue around it [129,130] (Figure 31 and Figure 32).

## 8. Neoplastic Diseases

HFUS is reported to determine tumoral depth and locoregional staging of skin neoplasms which is useful for treatment planning, particularly for melanoma, basal cell carcinoma (BCC), and squamous cell carcinoma (SCC) [131,132,133]. Moreover, HFUS can diagnose local tumor recurrence as para-cicatricial or satellite masses (less than 2 cm from the primary tumor), in-transit metastases (equal or more than 2 cm from the primary tumor), and nodal metastases, also when not visible or not palpable. UHFUS can better detect small epidermal and/or upper dermal lesions. In addition, tumor recurrence mimickers can be adequately assessed in patients imaged for suspected tumor relapse. Furthermore, ultrasound-guided fine needle aspiration cytology of small suspected local recurrent tumors allows precise performance of the sampling [134].

BCC and SCC are usually oval or band-like hypoechoic dermal and/or hypodermal vascularized structures; basal cell carcinoma commonly presents hyperechoic spots within the lesion, and a high-risk recurrent subtype lesion is suggested when seven or more hyperechoic spots are detected using a conventional linear probe at 15 MHz. These hyperechoic spots can be less visible or absent in superficial variants of BCC, and anechoic areas may be seen within adenoid cystic BCC variants. Bowen disease, or squamous cell carcinoma in situ, can be evaluated with UHFUS and shows a wavy epidermal surface, hypoechogenicity of the lower part of the epidermis, and underlying dermis hypervascularity [135,136,137,138] (Figure 33, Figure 34, Figure 35, Figure 36 and Figure 37).

Melanomas frequently are dermal hypoechoic vascularized fusiform-shaped structures with peripheral inflammatory signs, such as decreased dermal echogenicity and increased subcutaneous echogenicity; in situ, a melanoma’s regular hyperechoic band between the lesion and the dermis can be shown by UHFUS; in flat lesions a well-circumscribed fusiform millimeter-sized tissue can be evaluated using UHFUS; ulcerated lesions can show epidermal irregularities and discontinuity; satellite lesions are hypoechoic, oval, well-defined, and vascularized masses; exophytic lesions show hyperechoic epidermis with rough borders and hypoechoic tissue between the epidermis and dermis; moreover, ultrasound can detect local satellite recurrence or in-transit regional relapse; metastatic lymph nodes can show asymmetrical cortical thickening, hypoechoic or anechoic regions due to high cellularity, and chaotic cortical vascularity; furthermore, ultrasound can guide lymph node cytology or biopsy [139,140,141,142,143].

Merkel cell carcinoma presents as a hypoechoic richly vascularized dermal nodule that tends to invade subcutaneous tissue or deeper layers such as muscle or bone, frequently with posterior acoustic enhancement and epidermal thickening [144,145].

UHFUS allows the assessment of imaging characteristics of primary cutaneous lymphomas, helping reach an accurate diagnosis and treatment response evaluation. Heterogeneous hypoechoic nodules or pseudonodular lesions are more frequently reported in B-cell lymphomas while areas with dermal and/or hypodermal infiltration seem to be more commonly associated with T-cell lymphomas [146,147] (Figure 38); moreover, UHFUS can show epidermal thickness increases in patients with cutaneous T-cell lymphoma [148].

HFUS can provide clues for identifying cutaneous metastasis, as hypoechoic dermal and/or subcutaneous masses with ill-defined margins, irregular shape, and vascularization can be observed in aggressive lesions [149].

Cutaneous classic Kaposi sarcomas are usually hypoechoic, homogeneous, or heterogeneous and can range from avascular to hypervascular lesions. An avascular lesion can only require follow-up; however, lesions with internal vascular signals can require therapy as they are more prone to rapid clinical progression [150].

Pilomatrixomas, benign tumors derived from the hair matrix, and dermatofibrosarcoma protuberans, a locally aggressive malignant fibrous tumor, have strong HFUS-correlated findings. In fact, pilomatrixomas are usually well-defined dermal–hypodermal junction “target-type” lesions with an echogenic center, hyperechoic calcium spots within the nodule, and hypoechoic rim [151] (Figure 39).

Dermatofibrosarcoma protuberans is described as an ill-defined oval hypoechoic dermal lesion with a hyperechoic hypodermal portion with convex edges or tentacle-like projections, commonly vascularized on color Doppler. Recurrent dermatofibrosarcoma protuberans usually are oval or irregular-shaped, hypoechoic hypodermal masses [152,153] (Figure 40).

After biopsy and immunohistochemical analysis, preoperative sonography with subcutaneous infiltration identification can help diagnose pleomorphic dermal sarcoma instead of atypical fibroxanthoma, improving treatment strategy, as resection with 2 cm safety margins and lymph node sonography to rule out lymph node involvement is necessary for pleomorphic dermal sarcoma [154,155].

Cutaneous leiomyomatous tumors can be hypoechoic, pseudonodular, dermo-hypodermal structures with peripheral bundles, posterior acoustic reinforcement artifacts, and internal vascularity [156].

Infantile hemangiomas in the proliferative phase are hypervascularized hypoechoic masses with hyperechoic peripheral borders; the partial regression phase is characterized by less vascularity and mixed echogenicity; avascularity and hyperechogenicity are usually reported in the total regression phase sometimes associated with hypertrophic hypodermal lipodystrophy [157].

HFUS can help distinguish between dermatofibroma (or fibrous histiocytoma) high-risk and low-risk lesions; in fact, high-risk ones tend to be thicker with the subcutaneous component, irregularly shaped, heterogeneous, and vascularized. Therefore, HFUS can be very useful for clinical management, as high-risk dermatofibromas need extensive surgical resection; instead, cryotherapy or laser treatment could be performed in low-risk lesions [158].

Other lesions in which ultrasound is very useful thanks to very characteristic features are epidermal cysts, lipomas, schwannomas, and nodular fasciitis [71] (Figure 41, Figure 42, Figure 43, Figure 44 and Figure 45).

## 9. Aesthetic Medicine

Ultra-high-frequency ultrasound helps assess the surface area and length of fat protrusions at the dermal subcutaneous junction, as cellulite is characterized by a discontinuous dermal–subcutaneous interface with papillae adiposae due to the presence of fibrous bands holding the dermis to the fascia; moreover, ultrasound can assess the effectiveness of anti-cellulite therapies [159,160] (Figure 46).

The amount of dermal water decreases during the aging process, and proteoglycans accumulate in the papillary dermis that determines the appearance of the subepidermal low echogenic band, whose thickness increases linearly with age in sun-exposed areas like the forehead, zygomatic region, and dorsal aspect of the forearm; therefore, subepidermal low echogenic band thickness is a valuable marker of photoaging [161,162]

Aged skin reveals a thinner epidermis compared to younger skin, and HFUS can show epidermal thickening and increased upper dermis echogenicity three months after TCA-based anti-aging peeling [163,164]

Twenty-four weeks after nasolabial fold dermal hyaluronic injection, UHFUS can demonstrate increased dermal thickness and the percentage of wrinkle 3D fullness, providing valuable insight into the maintenance of results over time [165].

Temples, forehead, supraorbital region, glabella, tear trough, nasolabial groove, nose, lip, cheek, and chin facial filler materials can be well assessed using HFUS and UHFUS. In fact, hyaluronic acid, polyacrylamide, and lipofilling can show different ultrasound characteristics, echogenicity, fluidity, and blood flow signals. Anechoic well-defined pseudocystic deposits are typical for hyaluronic acid filler, whereas multiple thin strips of hypoechoic and anechoic areas, scattered in the hypoechoic loose subcutaneous tissue and other planes, may represent the spread and different injection techniques of hyaluronic acid fillers [166] (Figure 47 and Figure 48).

Anechoic pseudocystic structures that can contain echoes are suggestive of polyacrylamide. Iso-hypoechoic areas with some hyperechoic linear intervals are reported in transplanted fat. Hyperechoic deposits with posterior acoustic reverberance artifacts, also called “snowstorms”, are suggestive of silicone oil. Undiluted calcium hydroxyapatite is represented by hyperechoic areas with strong posterior shadowing artifacts; however, diluted presentations of calcium hydroxyapatite can present no posterior acoustic shadowing artifacts. Hybrid fillers that combine calcium hydroxyapatite and hyaluronic acid are injected in the subcutaneous tissue and on HFUS are similar to calcium hydroxyapatite ones, but with lower posterior acoustic shadowing according to the dilution and mixing with hyaluronic acid. Polycaprolactone shows hypoechoic deposits with multiple bright hyperechoic spots with mini comet tail artifacts. Polymethylmethacrylate presents hyperechoic deposits with hyperechoic spots and mini comet tail artifacts with posterior acoustic shadow artifacts [166,167,168,169,170] (Figure 49).

Facial and gluteal region microparticles of poly-l-lactic acid and carboxymethylcellulose hyperechoic deposits show posterior acoustic shadowing at the time of injection but not at follow-up two weeks later [169]. Silicone implants can be well-defined uniform anechoic structures (Figure 50).

HFUS can show dermal thickness increase and long-lasting correction of moderate to severe nasolabial folds at long-term follow-up following filler injection, possibly suggesting increases in collagen production after treatment [171,172,173,174].

In case of complications, inflammatory nodules show increased blood flow signals on color Doppler, with reduced echogenicity (Figure 51), and fat necrosis may manifest as an anechoic zone within subcutis. Some patients need additional filler injections after removing previous filling material and do not remember the exact type of material previously injected or inserted. HFUS can help distinguish these materials before re-treatment, helping avoid potential complications and identifying illegal fillers [175,176,177,178].

HFUS is a valuable tool for aesthetic injectors as it allows the visualization of needle tip, arteries, and veins, increasing safety and ameliorating patients’ outcomes. Vascular mapping using color Doppler can be performed before and after procedures in high-risk face areas to reduce adverse events such as ischemia or tissue necrosis [179] (Figure 52, Figure 53, Figure 54, Figure 55 and Figure 56).

In case of vascular occlusion, including external compression, ultrasound-guided hyaluronidase application can be performed to re-establish vascular flow (Figure 57 and Figure 58). Adjunctive use of self-administered 50% N_2_O could help obtain relief of pain and anxiety and potentially additional perfusion improvement. Moreover, ultrasound can help evaluate filler migration, inflammatory reactions, or infections [180,181,182,183,184].

A late complication of filler injection could be the appearance of palpable nodules, and HFUS helps distinguish between anechoic well-defined hyaluronic acid filler deposits and hypoechoic pseudonodules or areas suggestive of granulomas [185]. Moreover, fibrosis can demonstrate hyperechoic dermal and hypodermal areas with posterior acoustic shadowing [186]. Ultrasound can also guide intralesional drug injections of steroids, hyaluronidase, and 5-fluorouracil anti-metabolite in the case of difficult-to-treat foreign body granuloma reactions [187] (Figure 59, Figure 60 and Figure 61).

Granulomatous reaction secondary to fractional radiofrequency microneedling can show epidermal hypertrophy and dermal inflammation on ultrasound [188].

With regard to botulinum toxin applications, ultrasound can be performed before and during injection as muscle, tendons, and other structures can be easily visualized to improve efficacy and safety. Moreover, postinjection complications can be evaluated, such as lobular panniculitis, myositis, pseudoaneurysm, and asymmetric muscle contraction that can cause unwanted wrinkles or ptosis [189].

Finally, ultrasound can assess subcutaneous fat thickness and fat survival in gluteal fat grafting. The fat grafts appear as hypoechoic nodules with similar echogenicity compared to the subcutaneous tissue. Moreover, real-time use of ultrasound in gluteal fat transfer is becoming increasingly recommended to reduce complications, such as fat embolism, as ultrasound allows visualization of the transfer in the subcutaneous space, avoiding intramuscular injection [190,191,192,193,194,195]. Additionally, ultrasound can guide cannula positioning for liposuction [196].

## 10. Discussion

Dermatologists can evaluate the skin with different imaging modalities, and HFUS and UHFUS are increasingly being used as a complementary tool to clinical examination and dermoscopy to reach a correct diagnosis and to evaluate the best way to treat a disease or to evaluate treatment efficacy and duration.

In fact, HFUS’s abilities to evaluate cutaneous neoplasm size and depth well are useful for procedural planning before surgical resection or radiotherapy, particularly in delineating accurate tumor margins and in areas where clinical inspection and dermoscopy are insufficient [197], and HFUS’s capability to differentiate between a granuloma and a hyaluronic acid filler nodule is essential to decide the best treatment of an aesthetic procedure complication. If the clinical aspects of a cutaneous lesion are uncertain, ultrasound can contribute to diagnosing benign lesions, avoiding unnecessary surgery or advanced imaging studies such as MRI and CT, and planning specific topical or systemic treatments, which is obviously possible in pediatric patients, too [198].

The fast-growing literature about HFUS and UHFUS demonstrates that these imaging modalities are already part of clinical dermatology practice and that this trend will tend to increase during the next years and decades, thanks to the diffusion of these new ultrasound machines and clinical expertise. Some barriers to dermatologic ultrasound were reported, but ultrasound education will certainly foster expanding future application of ultrasound in clinical practice [199,200,201].

Ultrasound imaging in skin diseases can be performed by dermatologists or other medical doctors who work strictly with dermatologists, such us radiologists; the latter eventuality could be particularly probable in large dermatological institutions in which radiologists with dermatoradiology expertise could also offer other imaging modalities when needed, allowing better treatment and management of dermatology patients [202]. Moreover, ultrasound imaging utilization is growing among plastic surgeons in aesthetic medicine and reconstructive microsurgery fields [121]. Obviously, when a cell-level imaging of the superficial part of the dermis is necessary, reflectance confocal microscopy should be performed by dermatologists, in particular for the diagnosis of melanocytic lesions where it has been proven to increase the diagnostic accuracy when coupled with dermoscopy; in fact, this nearly histological resolution is not possible with ultrasound imaging [203,204].

Ultrasound is an easily accessible and relatively cheap imaging technique. Performance data reported in the literature for diagnosing skin cancer in adults have 100% sensitivity and variable specificity (73–93%) [205]. A very good performance could also be achieved when dermatologic ultrasound is performed in primary care, taking advantage of teledermatology and teleultrasound diagnosis (sensitivity, 100%; specificity, 97.8%) [206]. As regards morphea, increased subcutaneous tissue echogenicity and increased cutaneous blood flow are 100% sensitive and 100% specific for signs of activity in the lesion [207]. Referring diagnosis was correct in 73% of the lesions, and the addition of ultrasound increased correctness to 97%. In a study about localized skin lesions, ultrasound’s overall sensitivity was 99%, specificity was 100%, and statistical diagnostic certainty was 99%. Referring diagnosis was correct in 73% of the lesions, and the addition of ultrasound increased correctness to 97%. However, ultrasound cannot easily detect lesions that are epidermal only or that measure less than 0.1 mm in depth [208].

As regards future directions, the assessment of foreign body evaluation accuracy using high-frequency or ultra-high-frequency ultrasound compared to conventional linear probe ultrasonography could be performed to evaluate the necessity to use this imaging modality in patients who need accurate and quick surgical resection to avoid complications and permanent pain due to incomplete removal. Another use that will probably expand with ultrasound is avoiding the need for serial biopsies in inflammatory conditions such as morphea or lichen planus. Future studies on rare malignancies, such as rare melanoma subtypes, are warranted based on the results of the more frequent types of melanomas. Moreover, the cost-effectiveness of these imaging modalities could be investigated to try to decide if the time to implement them in everyday clinical practice has arrived. More in-depth evaluation of the effects of anti-aging therapies or anti-cellulite treatments could be conducted in the future, and future research to evaluate the usefulness of shear wave elastography could be performed. Furthermore, HFUS could be used to objectively assess human hypertrophic scar thickness variation after any surgical or invasive procedure such as botulinum toxin injections and, therefore, to evaluate its effect depending on different toxin concentrations [209].

## 11. Conclusions

HFUS and UHFUS are highly effective imaging modalities for skin evaluation, and their popularity in everyday clinical practice is constantly growing to assess skin diseases, plan procedures, monitor treatment efficacy and its duration, as well as to help diagnose and manage treatment complications such as in aesthetic procedures. In fact, they are already changing and will continue to modify dermatological everyday clinical practice as these new imaging modalities can be implemented to reach the best diagnosis, treatment, and management of skin diseases.

## Figures and Tables

**Figure 1 medicina-61-00220-f001:**
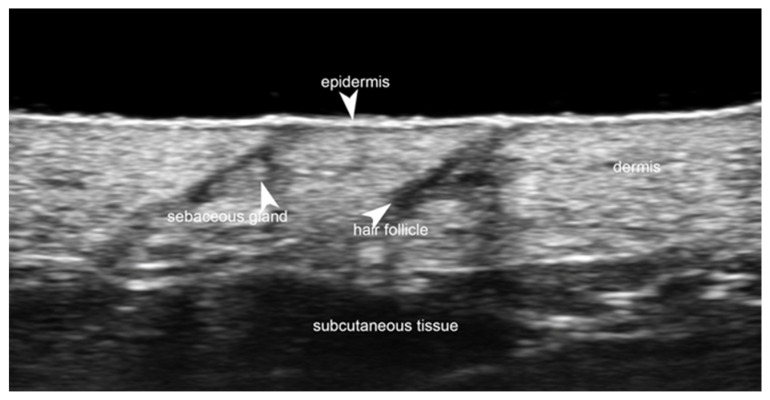
Normal non-glabrous skin at 70 MHz with a superficial bright hyperechoic monolaminar layer, which is the epidermis, a thicker but less hyperechoic band that corresponds to the dermis, and a deeper hypoechoic structure that represents the subcutaneous tissue. Notice the hyperechoic oval-shaped structure of the sebaceous gland and a hypoechoic oblique band that corresponds to the hair follicle.

**Figure 2 medicina-61-00220-f002:**
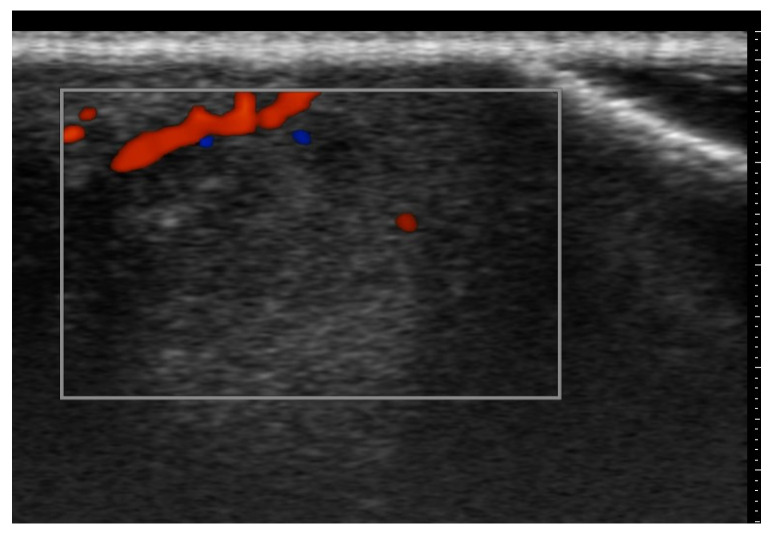
Third toe abscess in a color Doppler 70 MHz image: small inhomogeneous hypoechoic structure with peripheral hypervascularization.

**Figure 3 medicina-61-00220-f003:**
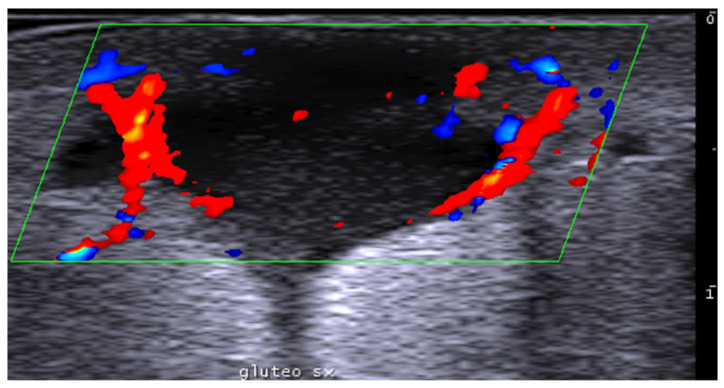
Abscess in a color Doppler 20 MHz image: hypoechoic fluid-filled structure with evident peripheral vascular signals.

**Figure 4 medicina-61-00220-f004:**
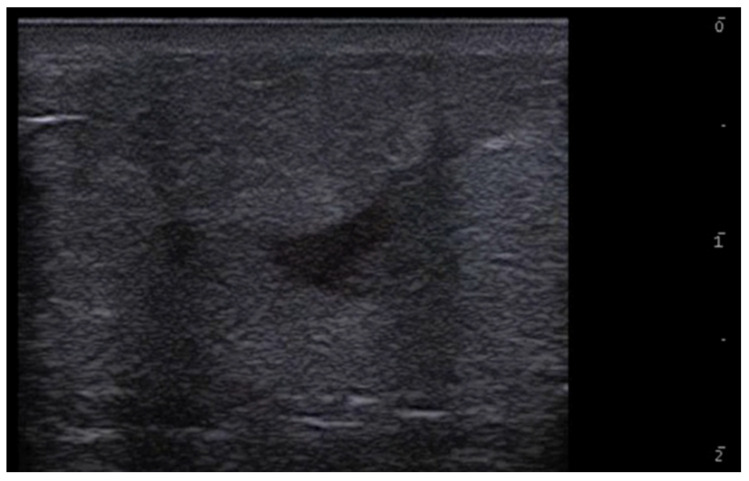
Grayscale 20 MHz ultrasound image of erysipelas shows a thickened, hyperechoic dermis with a hypoechoic band of fluid and hyperechoic surrounding subcutaneous edema, without abscess formation.

**Figure 5 medicina-61-00220-f005:**
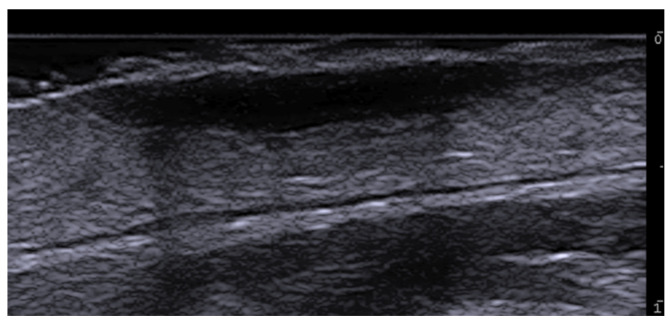
Grayscale 20 MHz ultrasound image of cutaneous leishmaniasis showing well-defined, homogenous hypoechoic dermal structure (between markers), with minimal reactive changes in the underlying hypodermis (pauci-inflammatory pattern).

**Figure 6 medicina-61-00220-f006:**
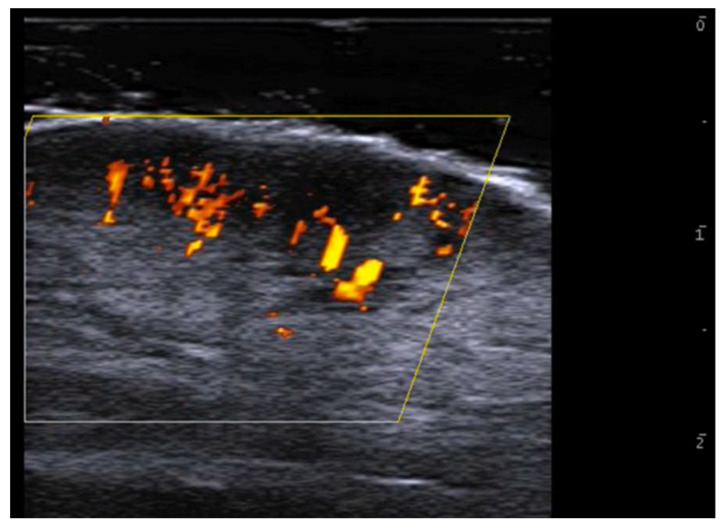
Power Doppler 20 MHz ultrasound image of an ulcerated plaque of cutaneous leishmaniasis shows a diffuse central vascular signal (inflammatory pattern).

**Figure 7 medicina-61-00220-f007:**
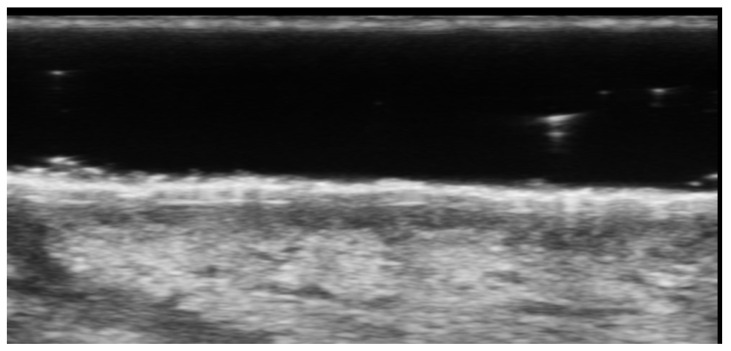
B-mode 70 MHz ultrasound image of atopic dermatitis with evident decreased echogenicity of the upper dermis.

**Figure 8 medicina-61-00220-f008:**
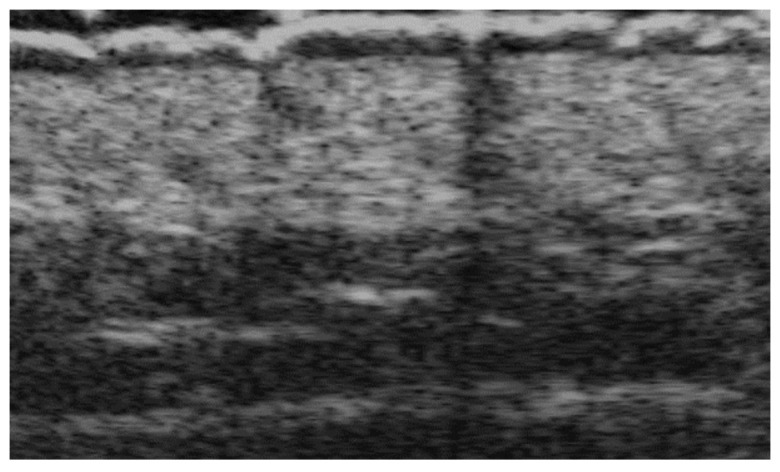
Atopic dermatitis with evident hypoechoic upper dermal band at 48 MHz.

**Figure 9 medicina-61-00220-f009:**
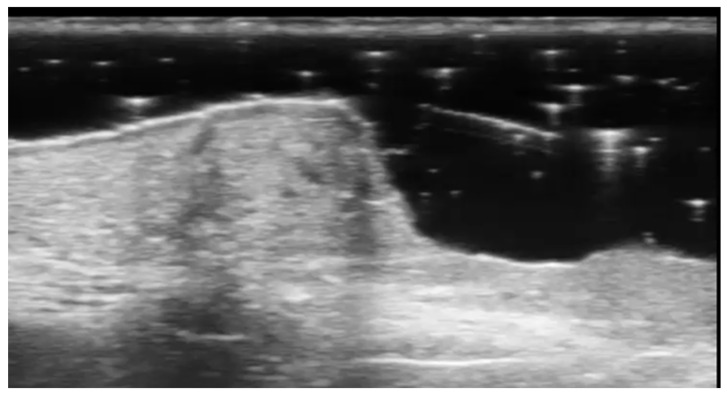
B-mode 70 MHz ultrasound image of a pyoderma gangrenosum lesion.

**Figure 10 medicina-61-00220-f010:**
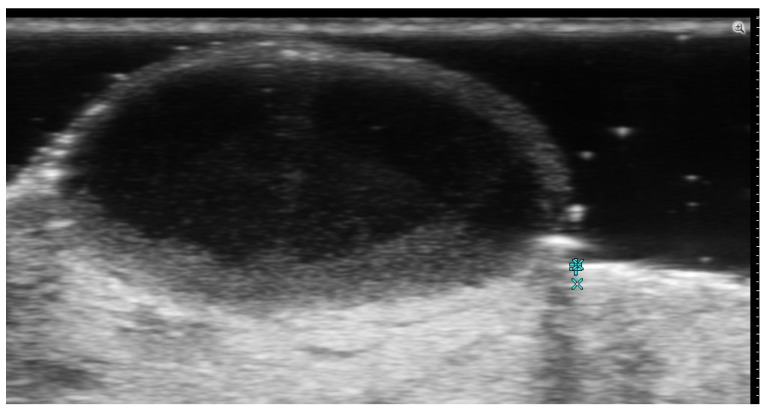
Subepidermal blister in bullous pemphigoid at 70 MHz.

**Figure 11 medicina-61-00220-f011:**
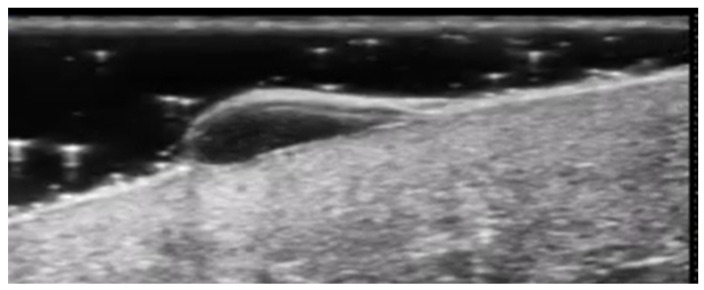
Intraepidermal blister in pemphigus vulgaris at 70 MHz.

**Figure 12 medicina-61-00220-f012:**
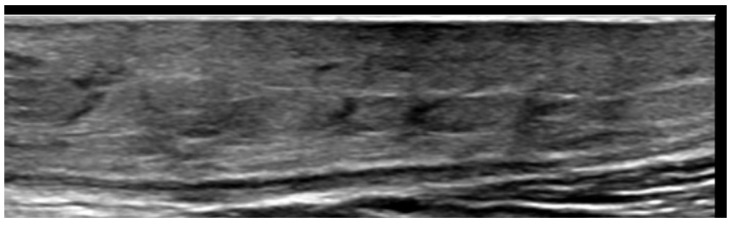
Panniculitis with hyperechoic lobules and hypoechoic septa in a thickened hypodermis using a 20 MHz probe.

**Figure 13 medicina-61-00220-f013:**
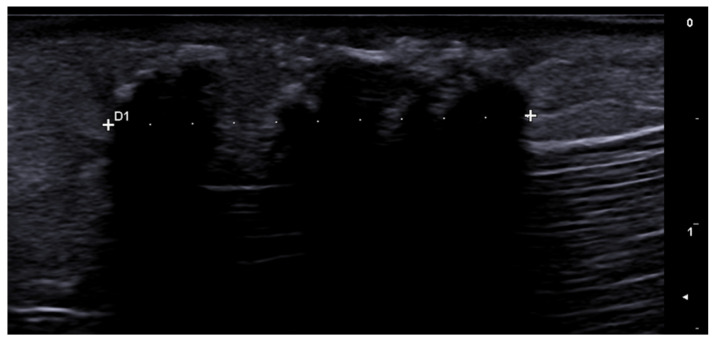
Right elbow subcutaneous calcifications in a patient with systemic sclerosis using a 24 MHz probe.

**Figure 14 medicina-61-00220-f014:**
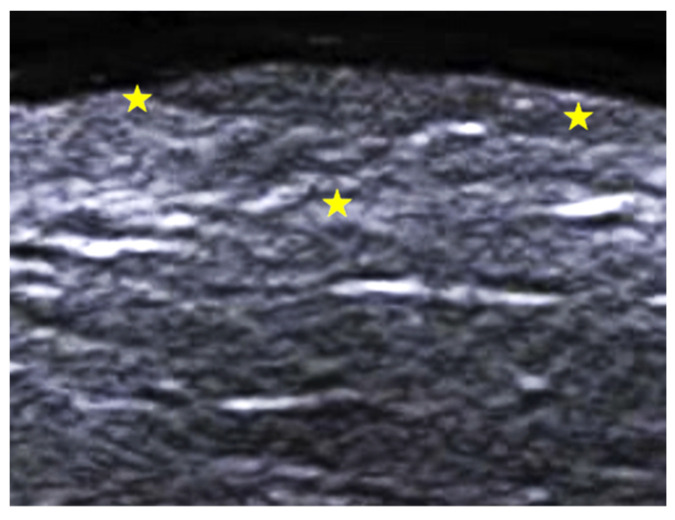
Grayscale 20 MHz ultrasound of pseudoxanthoma elasticum on the neck. The involved area is delimited by yellow stars and shows a mild decrease in dermal echogenicity in a small single papular lesion.

**Figure 15 medicina-61-00220-f015:**
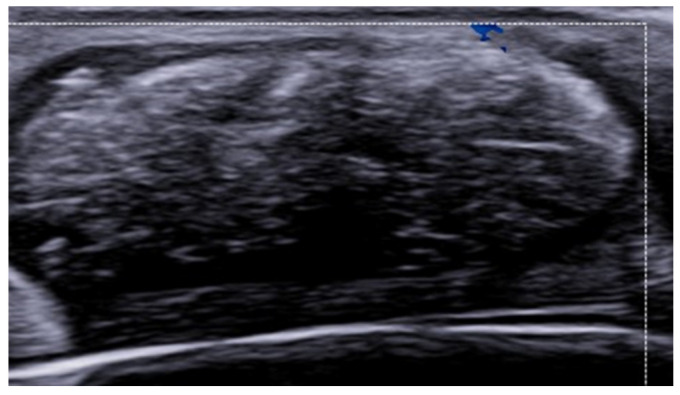
An oval dermal and subcutaneous heterogeneous hypoechogenic trichilemmal cyst in the scalp with keratinized material and hair fragment inside at 24 MHz.

**Figure 16 medicina-61-00220-f016:**
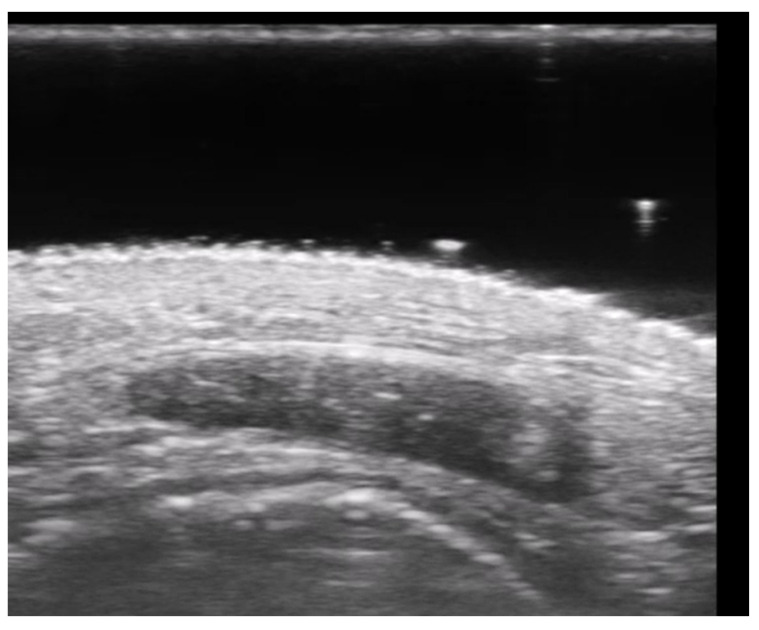
Acne pseudocyst at 70 MHz: oval-shaped hypoechoic structure within the hypodermis.

**Figure 17 medicina-61-00220-f017:**
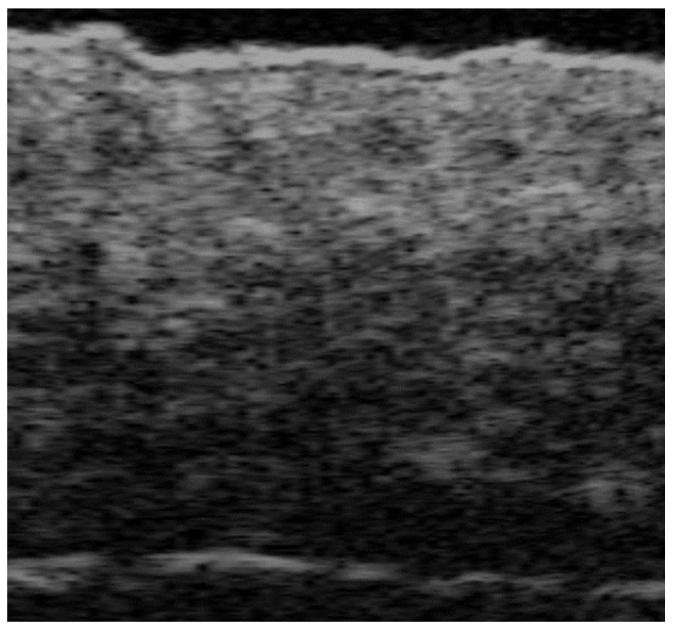
Forehead acne scar at 48 MHz with evident epidermal depression.

**Figure 18 medicina-61-00220-f018:**
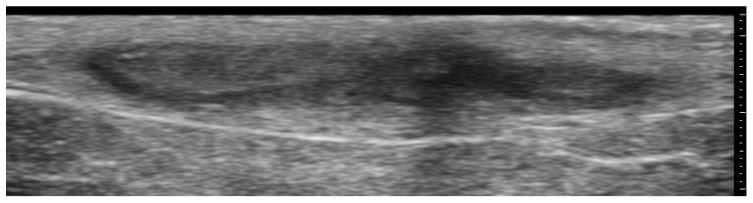
Hidradenitis suppurativa microtunnel at 70 MHz.

**Figure 19 medicina-61-00220-f019:**
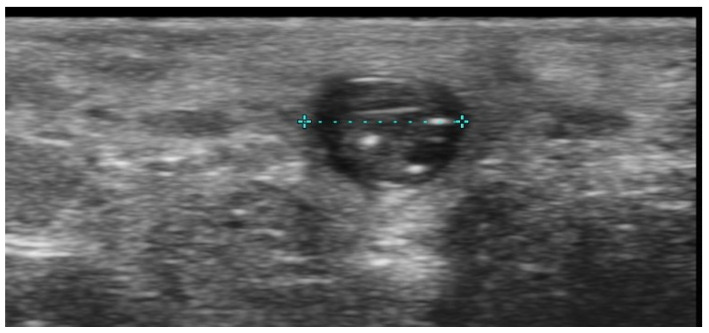
Hidradenitis suppurativa microcyst or ballooned hair follicle with 70 MHz probe.

**Figure 20 medicina-61-00220-f020:**
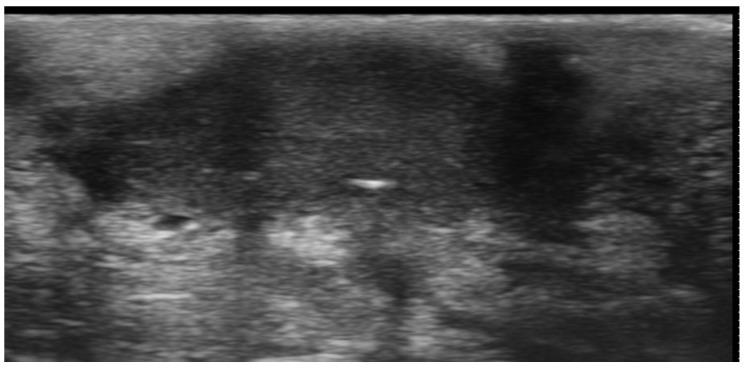
Hidradenitis suppurativa fluid collection at 70 MHz.

**Figure 21 medicina-61-00220-f021:**
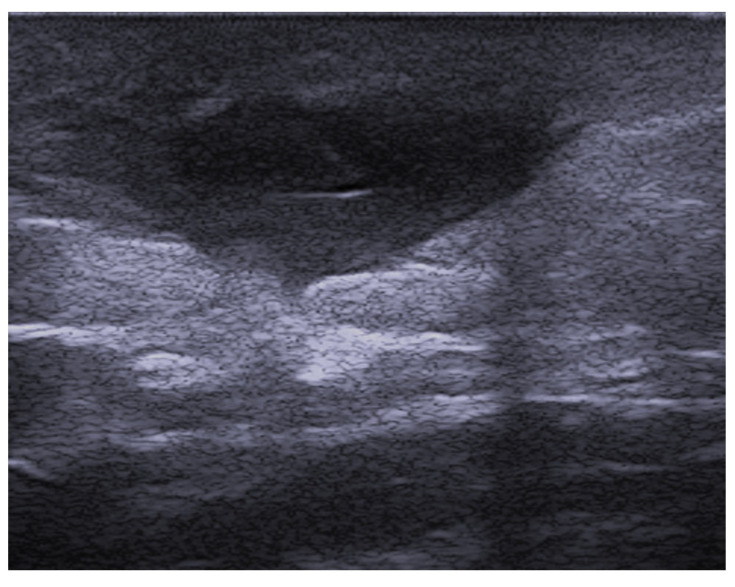
Grayscale 20 MHz ultrasound image showing a subcutaneous fluid collection in the buttock of a patient with hidradenitis suppurativa. A retained hair fragment is visible as a linear hyperechoic structure within the abscess and aligned parallel to the skin surface.

**Figure 22 medicina-61-00220-f022:**
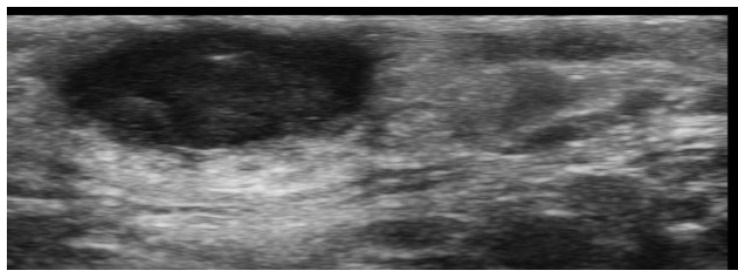
Hidradenitis suppurativa pseudocyst at 48 MHz.

**Figure 23 medicina-61-00220-f023:**
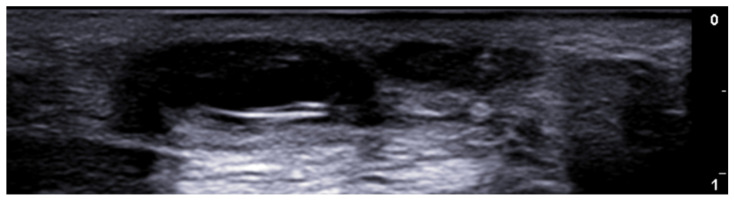
A 24 MHz ultrasound image showing a pilonidal sinus with hair fragments.

**Figure 24 medicina-61-00220-f024:**
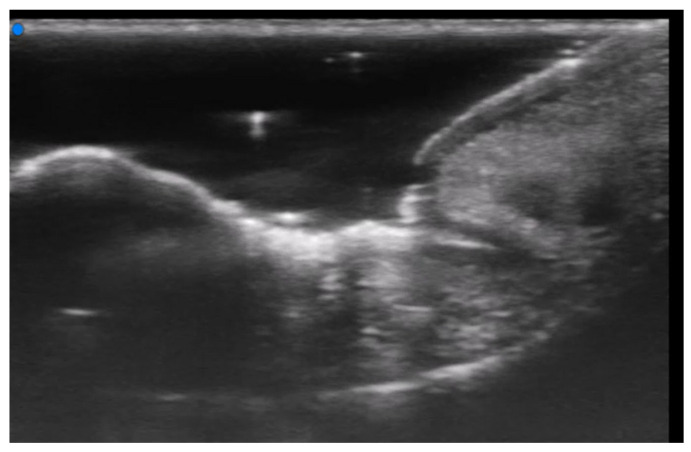
A 70 MHz ultrasound image shows a thickened, hyperechoic, and wavy nail plate with an increased thickness of the nail bed in a patient with psoriasis.

**Figure 25 medicina-61-00220-f025:**
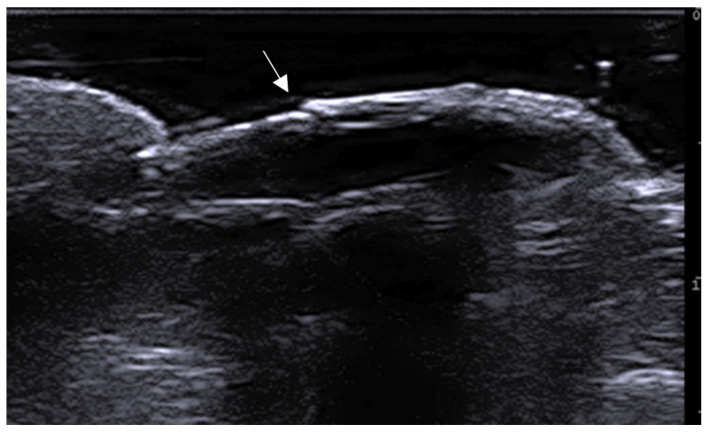
Longitudinal grayscale 20 MHz ultrasound of onychomadesis: note the abrupt interruption of the nail plate (arrow).

**Figure 26 medicina-61-00220-f026:**
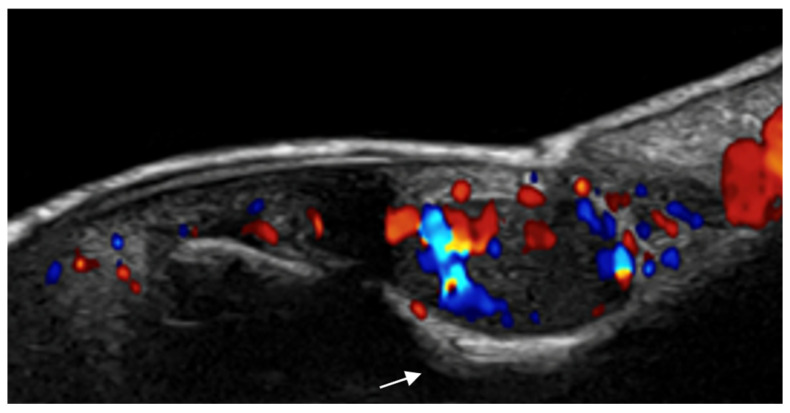
Glomus tumor of the right thumb at 24 MHz. Oval-shaped, well-defined, hypoechoic nodule that corresponds to a glomus tumor. There is hypervascularity within the tumor. Notice the scalloping of bony margin (arrow).

**Figure 27 medicina-61-00220-f027:**
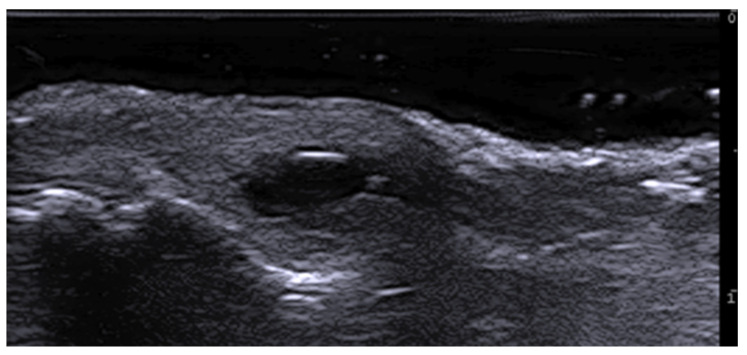
Longitudinal 20 MHz grayscale ultrasound of a digital myxoid cyst, showing a well-circumscribed anechoic lesion, with nail plate splitting due to compression on the matrix region and distal interphalangeal joint osteophytes.

**Figure 28 medicina-61-00220-f028:**
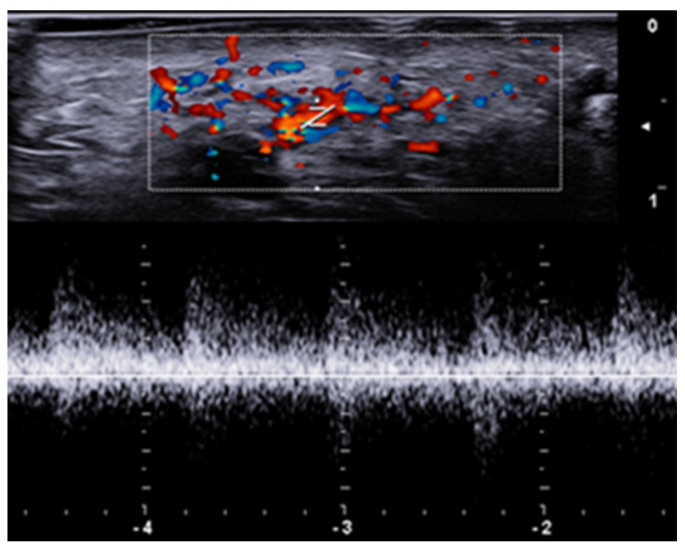
Left cheek vascular malformation with 24 MHz probe: hyperechoic subcutaneous tissue with evident vascular signals on color Doppler.

**Figure 29 medicina-61-00220-f029:**
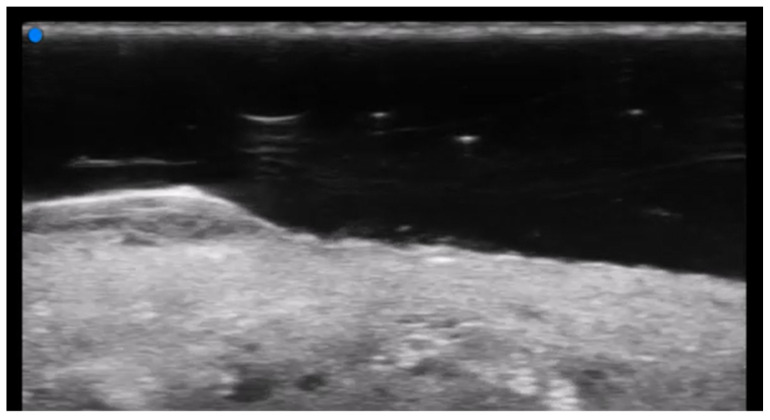
A 70 MHz ultrasound image shows a venous ulcer with a decreased echogenicity band in the perilesional tissue.

**Figure 30 medicina-61-00220-f030:**
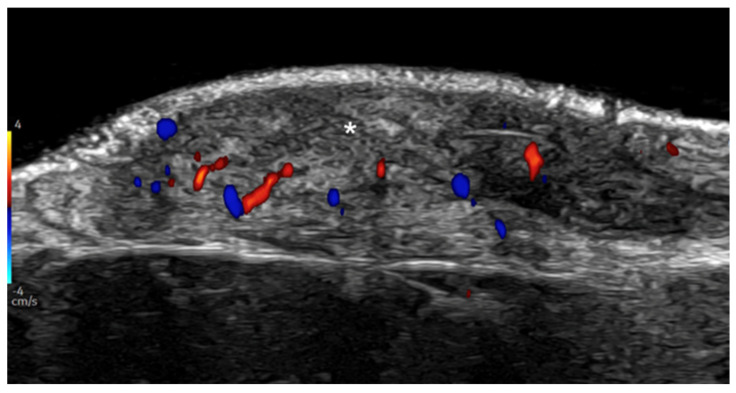
Active keloid. Hypoechoic laminar dermal thickening at 24 MHz that corresponds to a keloid (*). Notice the hypervascularity within the lesion.

**Figure 31 medicina-61-00220-f031:**
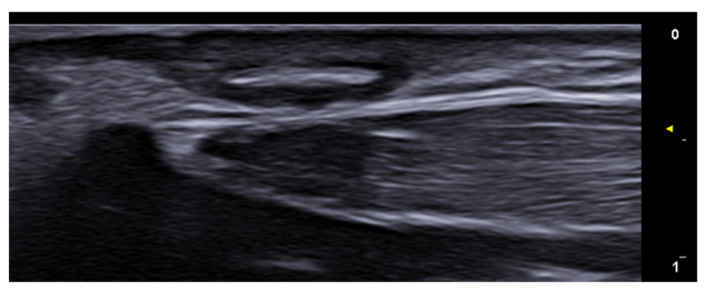
Garden cane fragment in the subcutis above the superficial head of first dorsal interosseous muscle of the right hand with a 24 MHz probe.

**Figure 32 medicina-61-00220-f032:**
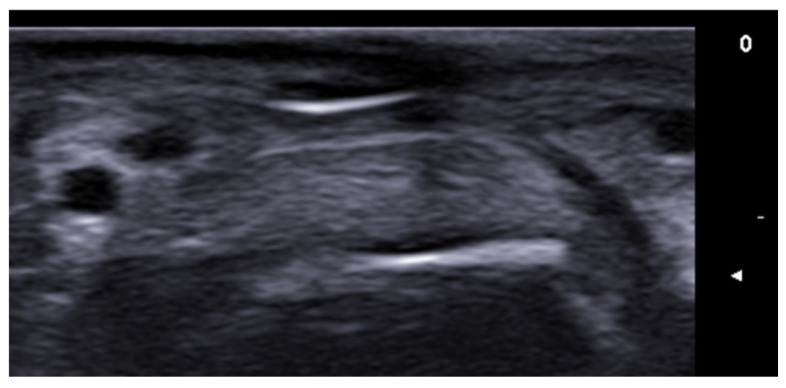
Glass fragment in the subcutis above the distal interphalangeal joint of the third finger with 24 MHz probe.

**Figure 33 medicina-61-00220-f033:**
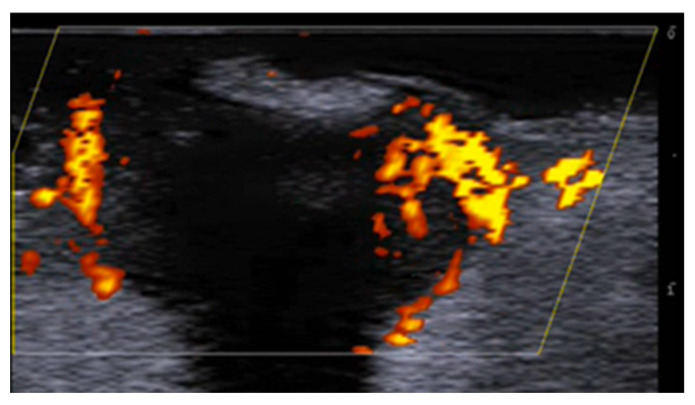
Power Doppler 20 MHz ultrasound image of an ulcerated nodular basal cell carcinoma of the preauricular area detects a well-defined, hypoechoic, round-shaped lesion with homogeneous echotexture, peripheral vascularity, and calcification with acoustic shadowing.

**Figure 34 medicina-61-00220-f034:**
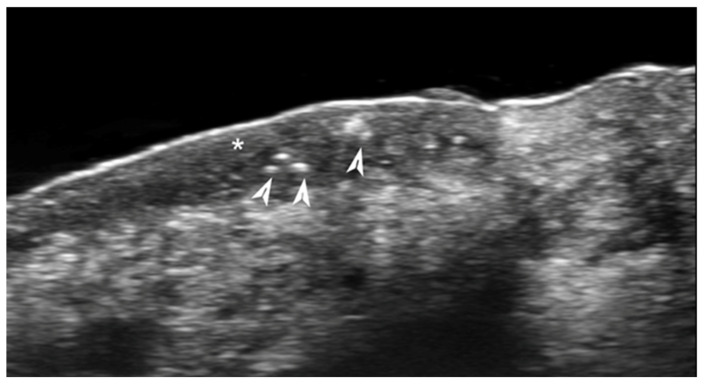
A 70 MHz ultra-high-frequency ultrasound image of a hypoechoic subepidermal and dermal basal cell carcinoma (*). Notice the hyperechoic spots within the tumor (arrows).

**Figure 35 medicina-61-00220-f035:**
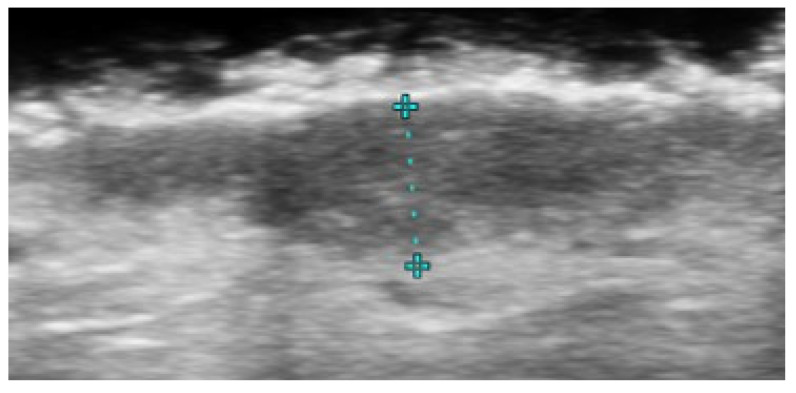
Hypoechoic subepidermal, dermal, and dermo-hypodermal junction cheek superficial basal cell carcinoma at 48 MHz.

**Figure 36 medicina-61-00220-f036:**
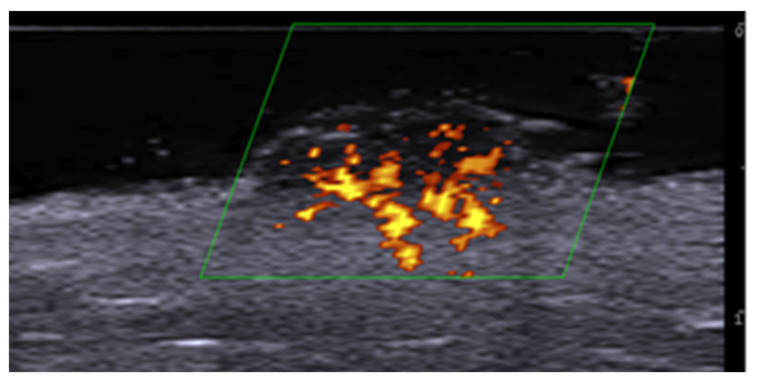
Power Doppler 20 MHz ultrasound image of an invasive G3 cutaneous squamous cell carcinoma reveals a hypoechoic, oval-shaped dermal mass with irregular borders, heterogeneous echotexture, and increased vascularity.

**Figure 37 medicina-61-00220-f037:**
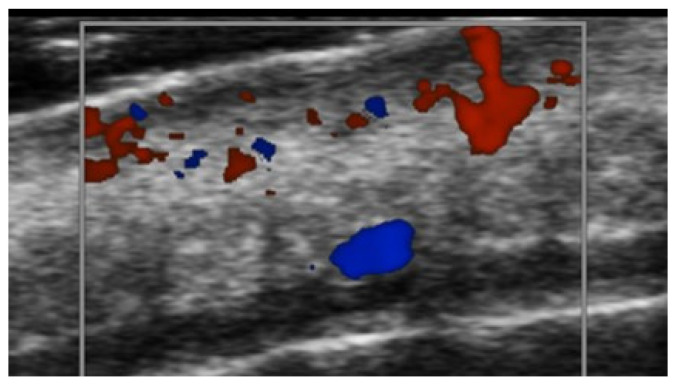
Color Doppler 48 MHz ultra-high-frequency ultrasound image of a hypoechoic in situ squamous cell carcinoma with mild mainly peripheral vascularization.

**Figure 38 medicina-61-00220-f038:**
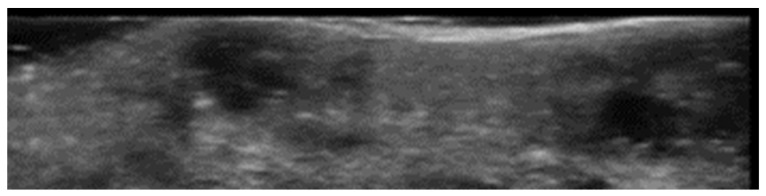
Two pseudonodular hypoechoic nose primary cutaneous B-cell lymphoma lesions at 48 MHz.

**Figure 39 medicina-61-00220-f039:**
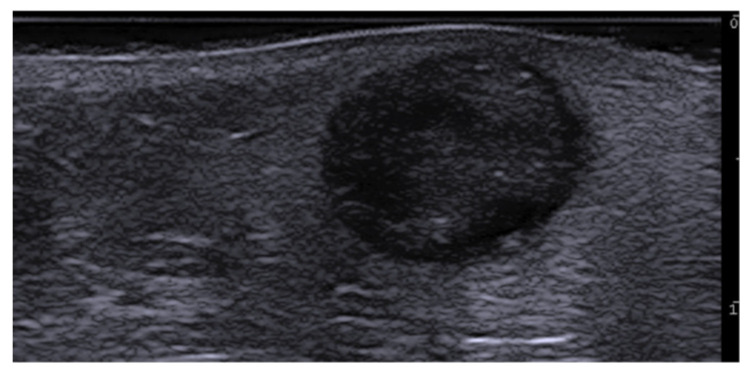
Grayscale 20 MHz ultrasound of a pilomatrixoma, showing a well-defined, round-shaped, mildly hypoechoic hypodermic mass with characteristic calcium spots and partial hypoechoic rim.

**Figure 40 medicina-61-00220-f040:**
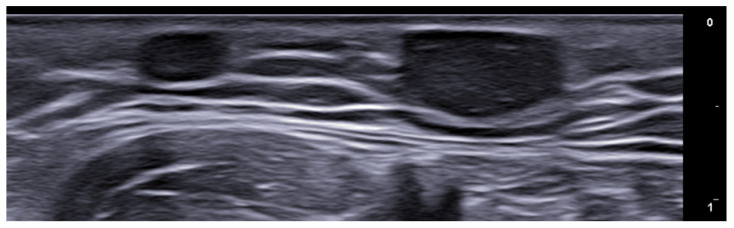
Subcutaneous dermatofibrosarcoma protuberans recurrent lesions with 24 MHz probe.

**Figure 41 medicina-61-00220-f041:**
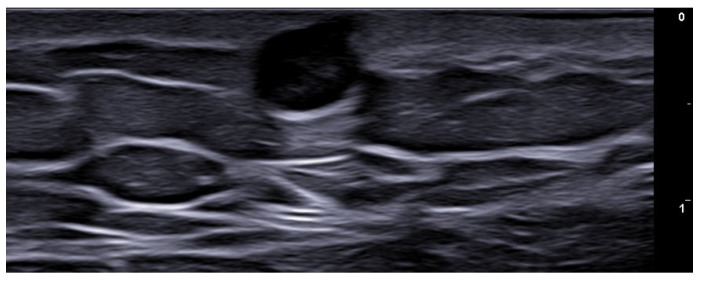
Dermo-epidermal hypoechoic round-shaped epidermal cyst with connecting tract toward the subepidermal and posterior enhancement at 24 MHz.

**Figure 42 medicina-61-00220-f042:**
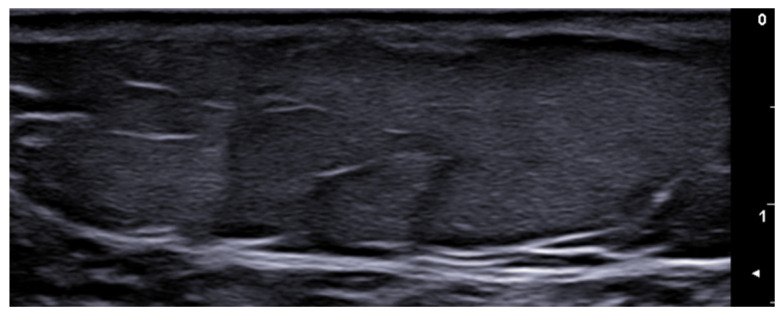
Mild hyperechoic oval-shaped lipoma of the left arm at 24 MHz.

**Figure 43 medicina-61-00220-f043:**
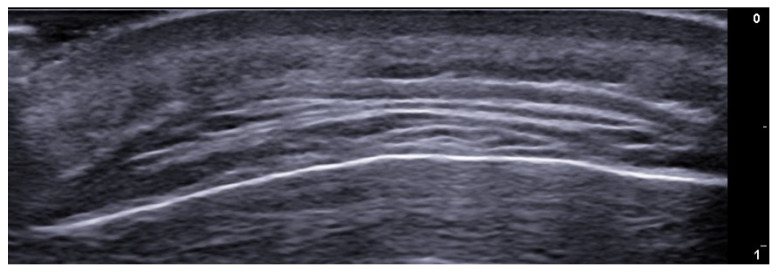
Well-defined hypoechoic subgaleal lipoma with hyperechoic septa between the frontalis muscle and the deep fascia at 24 MHz.

**Figure 44 medicina-61-00220-f044:**
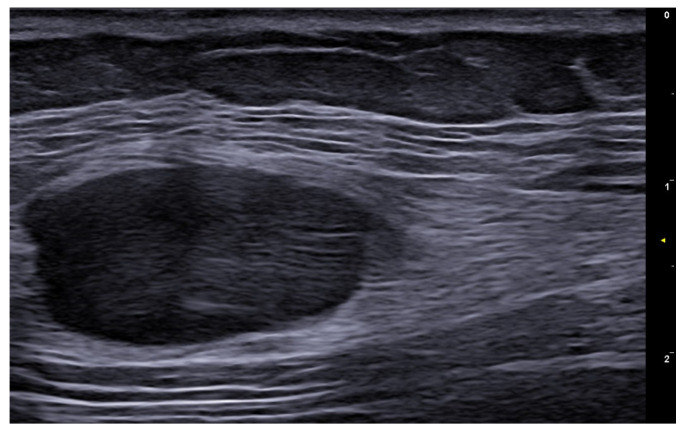
Well-delimited fusiform perifascial hypoechoic encapsulated schwannoma with mild posterior enhancement at 24 MHz.

**Figure 45 medicina-61-00220-f045:**
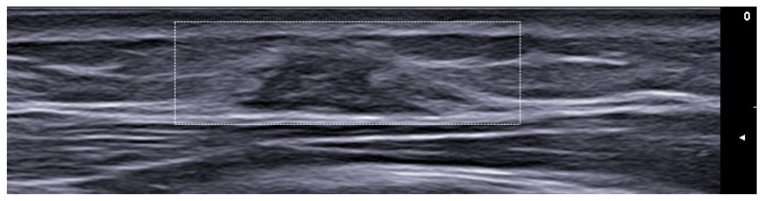
Irregular perifascial hypoechoic nodule with hyperechogenicity of the surrounding tissue because of edema, attached to the fascial layer in the deepest border representing nodular fasciitis at 24 MHz.

**Figure 46 medicina-61-00220-f046:**
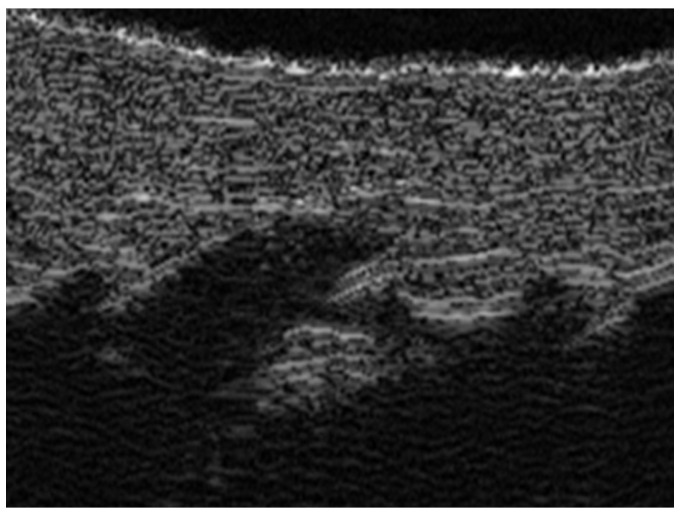
Cellulite at 50 MHz with evident papillae adiposae.

**Figure 47 medicina-61-00220-f047:**
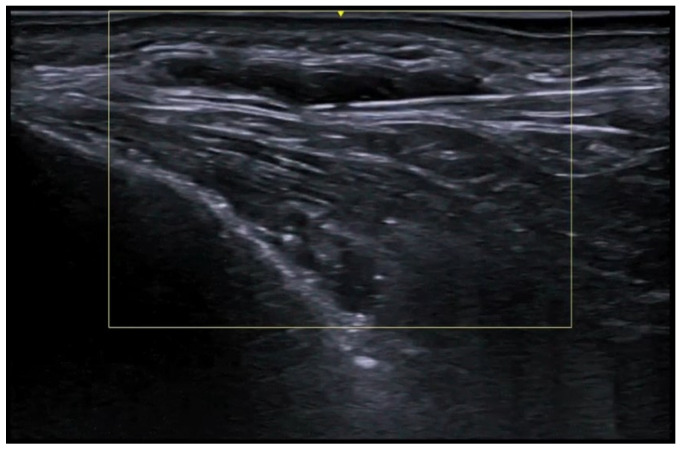
Temple hyaluronic acid filler injection at the interfascial plane (20 MHz).

**Figure 48 medicina-61-00220-f048:**
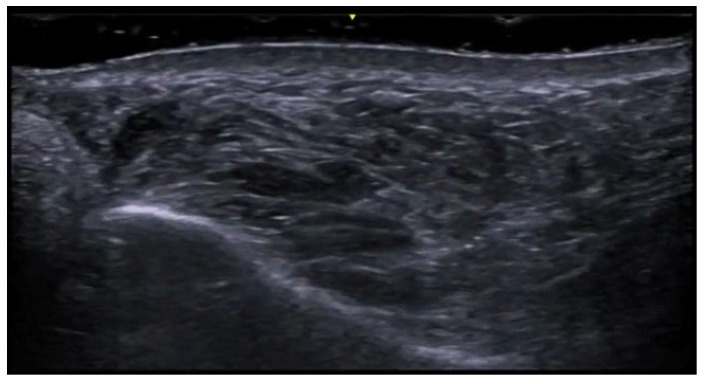
Suborbicularis oculi fat extensive deposits of hyaluronic acid, sagittal view at 20 MHz.

**Figure 49 medicina-61-00220-f049:**
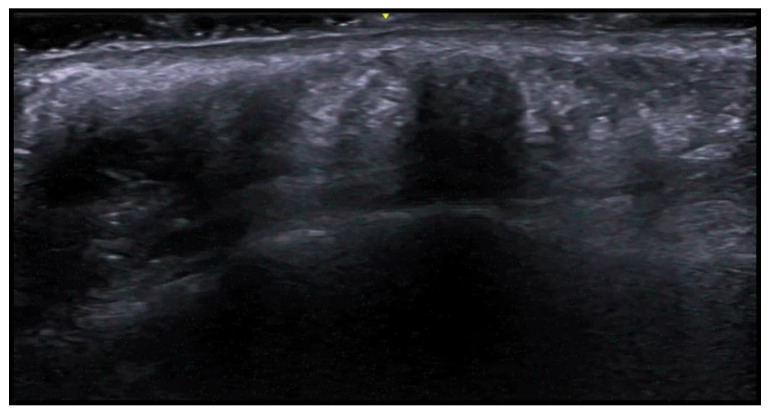
Prejowl sulcus polymethylmethacrylate at 20 MHz.

**Figure 50 medicina-61-00220-f050:**
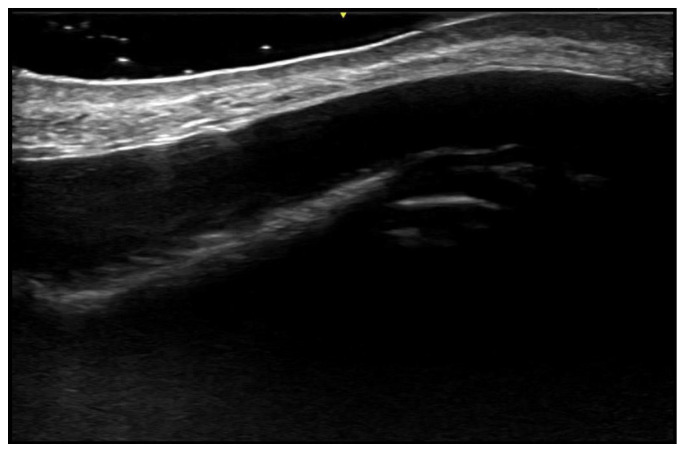
Nose silicone implant, sagittal view at 20 MHz.

**Figure 51 medicina-61-00220-f051:**
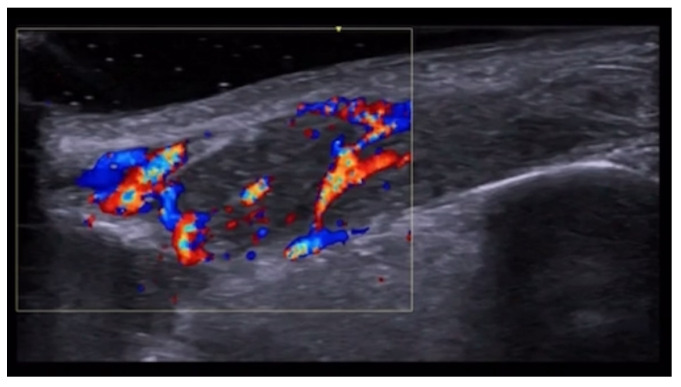
Color Doppler image of infraorbital inflammatory hyaluronic acid nodule at 20 MHz.

**Figure 52 medicina-61-00220-f052:**
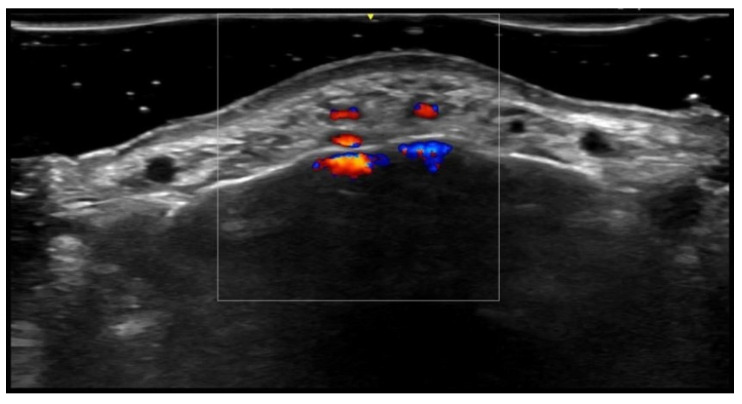
Color Doppler image of multiple glabella arteries and veins at 20 MHz.

**Figure 53 medicina-61-00220-f053:**
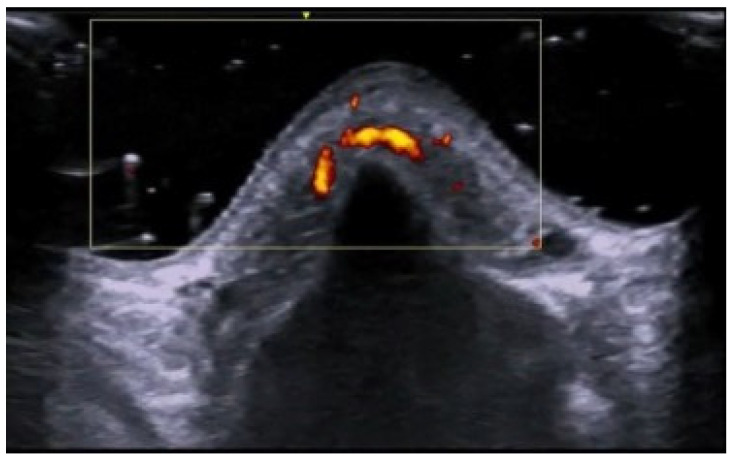
Color Doppler image of nasal radix intercanthal vein at 20 MHz.

**Figure 54 medicina-61-00220-f054:**
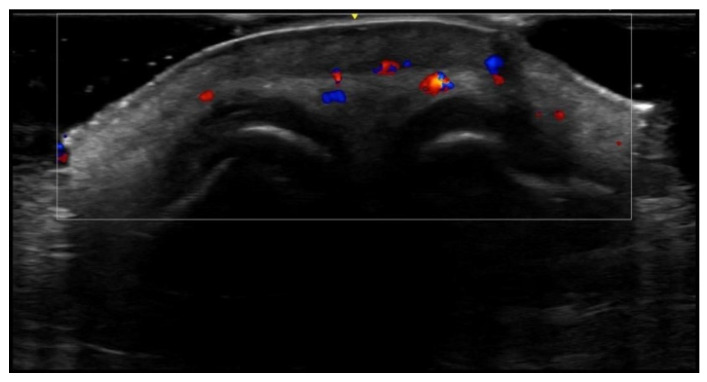
Color Doppler image of nasal tip cartilage and multiple arteries at 20 MHz.

**Figure 55 medicina-61-00220-f055:**
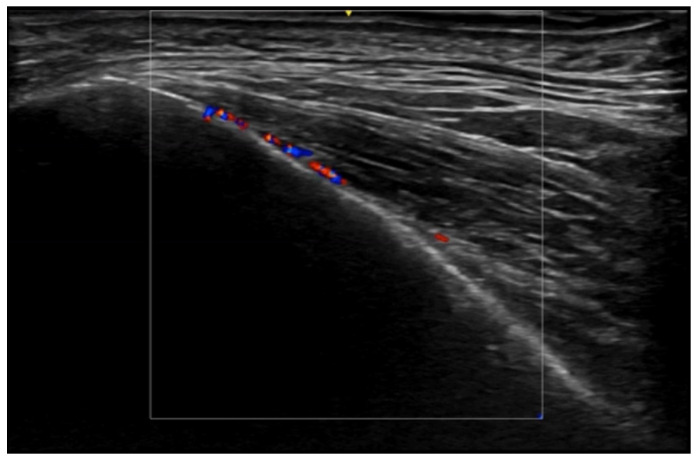
Color Doppler image of the deep temporal artery at 20 MHz.

**Figure 56 medicina-61-00220-f056:**
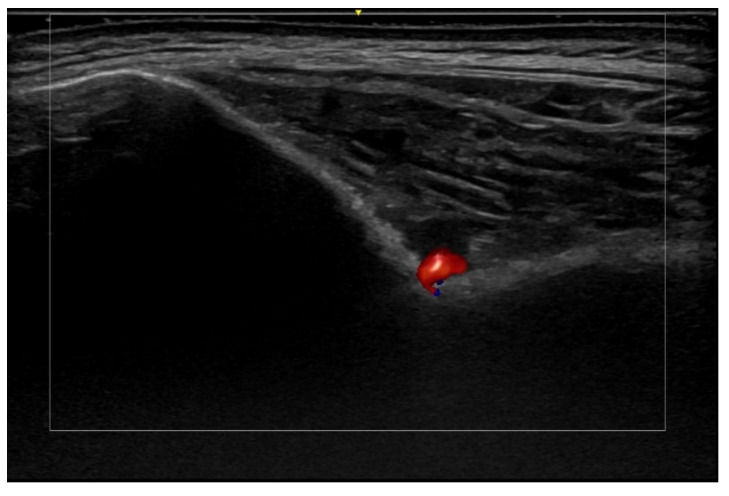
Color Doppler image of the zygomaticotemporal artery at 24 MHz.

**Figure 57 medicina-61-00220-f057:**
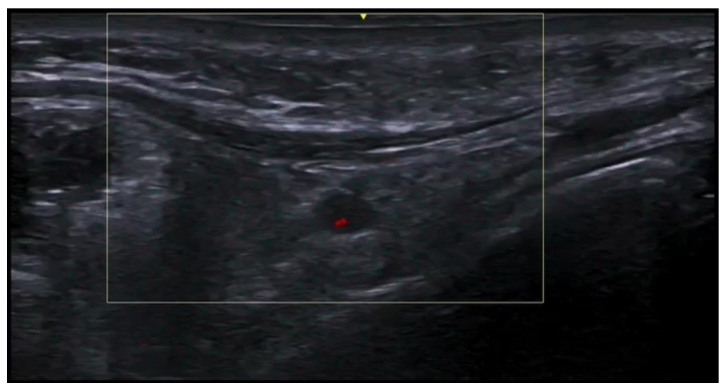
Facial artery near-occlusion at the mandible at 20 MHz.

**Figure 58 medicina-61-00220-f058:**
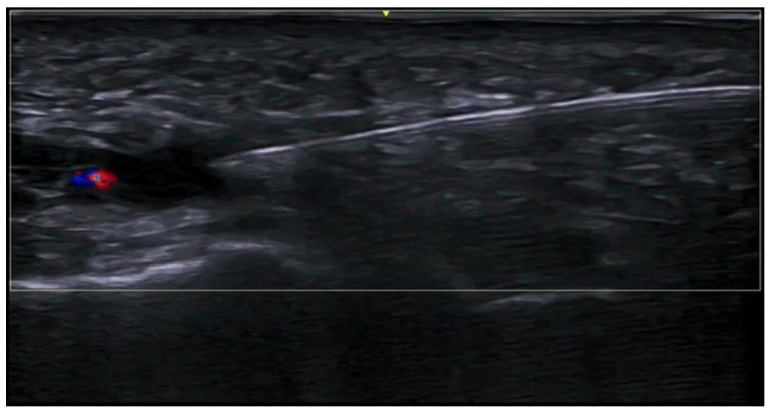
Angular artery with adjacent hyaluronic acid deposits: ultrasound-guided hyaluronidase injection in a case of vascular occlusion at 20 MHz.

**Figure 59 medicina-61-00220-f059:**
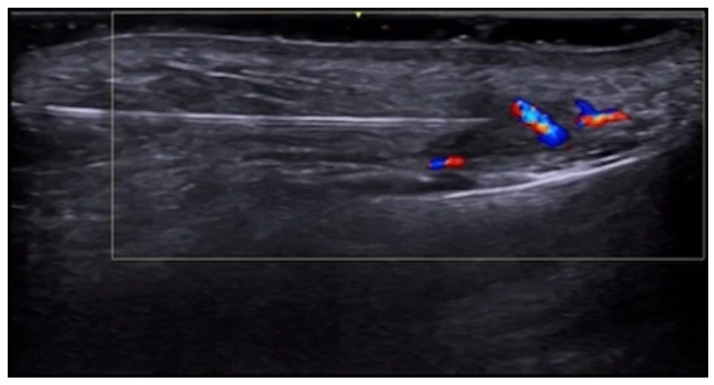
Ultrasound-guided hyaluronidase injection to an inflammatory infraorbital hyaluronic acid nodule.

**Figure 60 medicina-61-00220-f060:**
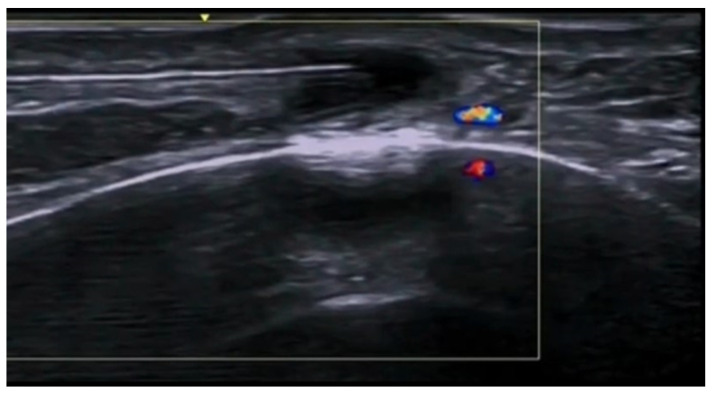
Hyaluronidase injection to an inflammatory prejowl hyaluronic acid nodule next to facial artery at 20 MHz.

**Figure 61 medicina-61-00220-f061:**
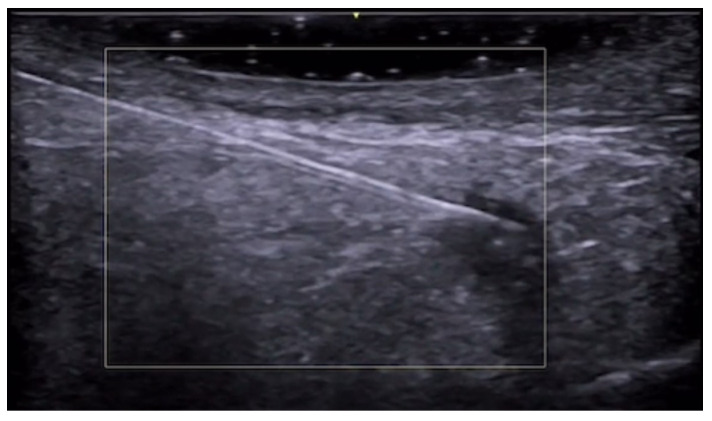
Hyaluronidase injection to retroseptal hyaluronic acid deposit at 20 MHz.

## Data Availability

Data are contained within the article.

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
