# Peer review of "High-Frequency and Ultra-High-Frequency Ultrasound in Dermatologic Diseases and Aesthetic Medicine"

_medicina, 2025, doi:10.3390/medicina61020220_

Round 1
Reviewer 1 Report
Comments and Suggestions for Authors
The article generally examines the diagnostic and therapeutic role of high-frequency and ultra-high-frequency ultrasound (HFUS and UHFUS) in dermatological diseases. It also focuses on the use of these technologies in aesthetic medicine. The article categorizes various dermatological conditions and details the ultrasound imaging features of each. In addition, images are added to support the explanations. However, we can say that the novelty of the article is limited since it only provides a literature review and does not provide any original experimental or clinical data. We can list the issues that need to be corrected as follows;
• It would be more appropriate if the original contributions of the study were stated in more detail and item by item in the introduction section.
• The introduction section does not provide a roadmap for the ongoing topics of the study. At the end of the introduction, add a short introductory sentence to the article sections.
• It would be appropriate to analyze the performance of HFUS and UHFUS compared to other diagnostic methods numerically. It would be useful to add tables comparing performance metrics such as sensitivity and specificity.
• It would be appropriate to compare HFUS/UHFUS with other imaging methods in terms of cost, accessibility and results. A comparison table indicating the strengths and weaknesses should be added.
• It explains in detail how HFUS and UHFUS are used in different dermatological conditions. In this respect, it can be said that it fills an important gap in the literature. However, I think that the information given under each heading should be better organized and interpreted in a clinical context. For example, it would be useful to add specific case examples on ultrasound-guided solutions to complications in aesthetic applications.
• The article is successful in showing the potential of HFUS and UHFUS in dermatological applications. However, I recommend that the authors add more examples from clinical practice and a section addressing the limitations of HFUS/UHFUS devices. In addition, short explanations can be added next to the images for the reader's easy understanding.
Author Response
The article generally examines the diagnostic and therapeutic role of high-frequency and ultra-high-frequency ultrasound (HFUS and UHFUS) in dermatological diseases. It also focuses on the use of these technologies in aesthetic medicine. The article categorizes various dermatological conditions and details the ultrasound imaging features of each. In addition, images are added to support the explanations. However, we can say that the novelty of the article is limited since it only provides a literature review and does not provide any original experimental or clinical data:
Thank you very much.
. It would be more appropriate if the original contributions of the study were stated in more detail and item by item in the introduction section:
Thank you very much.
. The introduction section does not provide a roadmap for the ongoing topics of the study. At the end of the introduction, add a short introductory sentence to the article sections:
Thank you very much. At the end of the introduction there is a sentence which introduce the article sections.
. It would be appropriate to analyze the performance of HFUS and UHFUS compared to other diagnostic methods numerically. It would be useful to add tables comparing performance metrics such as sensitivity and specificity:
Thank you very much. However, we think there are a lot of diseases, and it would be very difficult to summarize these performance data.
. It would be appropriate to compare HFUS/UHFUS with other imaging methods in terms of cost, accessibility and results. A comparison table indicating the strengths and weaknesses should be added;
Thank you very much for your suggestion. We added a sentence in the discussion about ultrasound limitation and when another imaging modality is necessary.
. It explains in detail how HFUS and UHFUS are used in different dermatological conditions. In this respect, it can be said that it fills an important gap in the literature. However, I think that the information given under each heading should be better organized and interpreted in a clinical context. For example, it would be useful to add specific case examples on ultrasound-guided solutions to complications in aesthetic applications:
Thank you very much for your suggestions. There are images about ultrasound-guided hyaluronidase injection.
. The article is successful in showing the potential of HFUS and UHFUS in dermatological applications. However, I recommend that the authors add more examples from clinical practice and a section addressing the limitations of HFUS/UHFUS devices. In addition, short explanations can be added next to the images for the reader's easy understanding:
Thank you very much for your suggestions. We added some ultrasound images explanations as suggested.
Reviewer 2 Report
Comments and Suggestions for Authors
i read with great interest the manuscript
it is indeed very nice and has a variety of images however it resembles more of a chapter in a book- rather than a review manuscript
i would suggest combining some of the ultrasound images with the actual macroscopic or dermoscopy images( if exist) and commenting their assotiation- this would be very nice
a comment on the ultrasound and histopathology of the lesion should be added too in discussion
also a table summarising the ultrasound findings in each cateagory ( infections, inflammatory etc) would increase the readability
some ultra images do not have detailed descriptions such as abscesses
All in all a very informative paper but needed of certain improvements
Author Response
-it is indeed very nice and has a variety of images however it resembles more of a chapter in a book- rather than a review manuscript:
Thank you very much.
-i would suggest combining some of the ultrasound images with the actual macroscopic or dermoscopy images (if exist) and commenting their association- this would be very nice:
Thank you very much for your suggestion; however, we prefer to insert ultrasound images only; we think there are a lot of images yet, and moreover we do not have the majority of associated macroscopic or dermoscopy images.
-a comment on the ultrasound and histopathology of the lesion should be added too in discussion:
Thank you very much for this suggestion; however, there are a lot of diseases, so it could be very difficult for us to choose to comment only few diseases.
-also a table summarising the ultrasound findings in each category (infections, inflammatory etc) would increase the readability:
Thank you very much for your suggestion; however, we do not think that is simple to summarize in a table something that is written is the text in few sentences; we do not think it could help so much.
-some ultra images do not have detailed descriptions such as abscesses:
thank you very much for this suggestion: we added some ultrasound images explanations as suggested.
-All in all a very informative paper but needed of certain improvements:
Thank you very much.
Round 2
Reviewer 1 Report
Comments and Suggestions for Authors
The article generally examines the diagnostic and therapeutic role of high-frequency and ultra-high-frequency ultrasound (HFUS and UHFUS) in dermatological diseases. It also focuses on the use of these technologies in aesthetic medicine. The article categorizes various dermatological conditions and details the ultrasound imaging features of each. In addition, images are added to support the explanations. However, we can say that the novelty of the article is limited since it only provides a literature review and does not provide any original experimental or clinical data.
It is seen that most of the issues stated for correction have not been carried out by the authors. It has been concluded that the article does not meet the qualifications of the article in its current state.
Author Response
Thank you very much for your comments and suggestions.
. The article generally examines the diagnostic and therapeutic role of high-frequency and ultra-high-frequency ultrasound (HFUS and UHFUS) in dermatological diseases. It also focuses on the use of these technologies in aesthetic medicine. The article categorizes various dermatological conditions and details the ultrasound imaging features of each. In addition, images are added to support the explanations. However, we can say that the novelty of the article is limited since it only provides a literature review and does not provide any original experimental or clinical data:
Thank you very much.
Obviously in a review, as stated in https://www.mdpi.com/journal/diagnostics/instructions, no new, unpublished data should be presented.
The novelty of the article is that we extensively evaluated HFUS and UHFUS in infections, inflammatory dermatoses, metabolic and genetic disorders, specific cutaneous structure skin disorders, vascular and external agents-associated disorders, neoplastic diseases, and aesthetics.
We added the term extensively in the manuscript.
As you know better than us, the literature change quickly, so every review will add something to previous articles, particularly in the fast-growing literature about ultrasound in dermatology.
. It would be more appropriate if the original contributions of the study were stated in more detail and item by item in the introduction section:
Thank you very much. What do you mean about original contribution?
We added the term extensively in the manuscript. We thing we did an extensive work, I did not find anything similar in the literature.
. The introduction section does not provide a roadmap for the ongoing topics of the study. At the end of the introduction, add a short introductory sentence to the article sections:
Thank you very much. At the end of the introduction there is a sentence which introduce the article sections. “In this narrative review, we evaluated extensively HFUS and UHFUS in infections, inflammatory dermatoses, metabolic and genetic disorders, specific cutaneous structure skin disorders, vascular and external agents-associated disorders, neoplastic diseases, and aesthetics”. We think this sentence introduce the following article sections.
. It would be appropriate to analyze the performance of HFUS and UHFUS compared to other diagnostic methods numerically. It would be useful to add tables comparing performance metrics such as sensitivity and specificity:
Thank you very much. However, the data about sensitivity and specificity are scarce in the literature, we added the few data that we found in the discussion. If you have more information than us about that let us know and we could write better.
We added these sentences: ”Performance data reported in the literature for diagnosing skin cancer in adults are 100% sensitivity and variable specificity (73-93%) [238]. A very good performance could also be achieved when dermatologic ultrasound is performed in primary care taking ad-vantage of teledermatology and teleultrasound diagnosis (sensitivity, 100%; specificity, 97.8%) [239]. As regards morphea, increased subcutaneous tissue echogenicity and in-creased cutaneous blood flow are 100% sensitive and 100% specific for signs of activity in the lesion [240]. Referring diagnosis was correct in 73% of the lesions, and addition of ultrasound increased correctness to 97%. In a study about localized skin lesions, ultrasound overall sensitivity was 99%, specificity was 100%, and statistical diagnostic certainty was 99%. Referring diagnosis was correct in 73% of the lesions, and addition of ultrasound increased correctness to 97%. However, ultrasound cannot detect easily lesions that are epidermal only or that measure less than 0.1 mm in depth [241, 242].”
. It would be appropriate to compare HFUS/UHFUS with other imaging methods in terms of cost, accessibility and results. A comparison table indicating the strengths and weaknesses should be added;
Thank you very much for your suggestion. We added a sentence in the discussion about ultrasound limitation and when another imaging modality is necessary.
We do not found data about cost-effectiveness in the literature. Obviously, the cost of reflectance confocal microscopy is higher, but the disease targets are different, so we think that a comparison is not appropriate, and as previously written, we did not find a previous previously published article. Ultrasound is obviously easily accessible. We added a sentence in the discussion: “Ultrasound is an easily accessible and relatively cheap imaging technique”
. It explains in detail how HFUS and UHFUS are used in different dermatological conditions. In this respect, it can be said that it fills an important gap in the literature. However, I think that the information given under each heading should be better organized and interpreted in a clinical context. For example, it would be useful to add specific case examples on ultrasound-guided solutions to complications in aesthetic applications:
Thank you very much for your suggestions. There are images about ultrasound-guided hyaluronidase injection.
. The article is successful in showing the potential of HFUS and UHFUS in dermatological applications. However, I recommend that the authors add more examples from clinical practice and a section addressing the limitations of HFUS/UHFUS devices. In addition, short explanations can be added next to the images for the reader's easy understanding:
Thank you very much for your suggestions. We previously added some ultrasound images explanations as suggested. We previously added a sentence in the discussion about ultrasound limitation and when another imaging modality is necessary. We added this sentence, too: “However, ultrasound cannot detect easily lesions that are epidermal only or that measure less than 0.1 mm in depth”
Reviewer 2 Report
Comments and Suggestions for Authors
although the authors did not deal satisfactorily with my suggestions the manuscript remains informative
Author Response
Thank you very much for everything.